# Molecular determinants of large cargo transport into the nucleus

Giulia Paci[1,2,3], Tiantian Zheng[4], Joana Caria[1,2,3], Anton Zilman[4,5]*, Edward A Lemke[1,2,3]*

[1]Biocentre, Johannes Gutenberg-University Mainz, Mainz, Germany; [2]Institute of Molecular Biology, Mainz, Germany; [3]European Molecular Biology Laboratory, Heidelberg, Germany; [4]Department of Physics, University of Toronto, Toronto, Canada; [5]Institute for Biomaterials and Biomedical Engineering (IBBME), University of Toronto, Toronto, Canada

**Abstract** Nucleocytoplasmic transport is tightly regulated by the nuclear pore complex (NPC). Among the thousands of molecules that cross the NPC, even very large (>15 nm) cargoes such as pathogens, mRNAs and pre-ribosomes can pass the NPC intact. For these cargoes, there is little quantitative understanding of the requirements for their nuclear import, especially the role of multivalent binding to transport receptors via nuclear localisation sequences (NLSs) and the effect of size on import efficiency. Here, we assayed nuclear import kinetics of 30 large cargo models based on four capsid-like particles in the size range of 17–36 nm, with tuneable numbers of up to 240 NLSs. We show that the requirements for nuclear transport can be recapitulated by a simple two-parameter biophysical model that correlates the import flux with the energetics of large cargo transport through the NPC. Together, our results reveal key molecular determinants of large cargo import in cells.

**\*For correspondence:**
zilmana@physics.utoronto.ca (AZ);
edlemke@uni-mainz.de (EAL)

**Competing interests:** The authors declare that no competing interests exist.

## Introduction

Cargo transport across the nuclear envelope is a hallmark of eukaryotic cells and is central to cellular viability (*Knockenhauer and Schwartz, 2016*; *Jamali et al., 2011*; *Fahrenkrog and Aebi, 2003*). In a typical HeLa cell, more than 2000 nuclear pore complexes (NPCs) span the nuclear envelope (*Ribbeck and Görlich, 2001*; *Maul et al., 1972*). With ≈120 MDa in metazoans (*Reichelt et al., 1990*) and roughly half that weight in yeast (*Rout and Blobel, 1993*; *Yang et al., 1998*), the NPC is among the largest macromolecular complexes found inside the cell. NPCs are the gatekeepers of nucleocytoplasmic transport and restrict access of cargoes larger than the typically reported threshold of 40 kDa (*Paine et al., 1975*; *Keminer and Peters, 1999*; *Mohr et al., 2009*), albeit recent work points to a rather 'soft' barrier model and a gradual decrease of passive transport rates with size (*Timney et al., 2016*). However, cargoes that present a nuclear localisation sequence (NLS) and bind nuclear transport receptors (NTRs) can rapidly enter into the nucleus. Several studies have characterized the NTR-mediated transport process, typically focusing on cargoes with one to five NLSs, and their nuclear import kinetics have been shown to follow a mono-exponential behaviour (*Ribbeck and Görlich, 2001*; *Kopito and Elbaum, 2007*; *Timney et al., 2006*).

NPCs are remarkable in the diversity of sizes of cargoes they can transport, ranging from import of nuclear proteins (including histones and transcription factors), to viral import and nuclear export of pre-ribosomal subunits and mRNA complexes (*Panté and Kann, 2002*; *Grünwald and Singer, 2010*; *Grünwald et al., 2011*; *Babcock et al., 2004*; *Mor et al., 2010*; *Au and Panté, 2012*). How very large cargoes (>15 nm) can be efficiently transported is still an enigma, especially considering the dimensions and structure of the transport conduit itself. The NPC is formed by multiple copies of about 30 proteins, two thirds of which are folded proteins that assemble the NPC scaffold. The

**eLife digest** Eukaryotes, such as animals, plants and fungi, store the genetic material within their cells inside a specific compartment called the nucleus. Surrounding the nucleus is a protective membrane which molecules must pass across in order to reach the cell's DNA. Straddling the membrane are nuclear pore complexes, or NPCs for short, which act as the gatekeepers to the nucleus, shuttling thousands of different molecules back and forth whilst restricting access to others.

Large cargoes need to have specific markers on their surface called nuclear localization signals in order to be transported by NPCs. Certain transporter proteins help the NPC carry large molecules across the membrane by binding to these signals. This generates the energy needed to overcome the barrier of transporting it across the membrane.

Some viruses have nuclear localization signals of their own, which can exploit this transport system; these signals allow the virus to enter the nucleus and hijack the genetic machinery of the cell. It has been suggested that viruses have multiple copies of these surface signals to improve their chances of reaching the nucleus. However, it remained unclear how the number of nuclear localization signals affects the transport of large molecules into the nucleus.

To answer this question, Paci et al. engineered a range of different sized particles derived from viral structures which had varying numbers of nuclear localization signals on their surface. These particles were inserted into human cell lines grown in the laboratory, and imaged to see how they were transported into the nucleus. The rate of nuclear transport was then measured for each particle, and this data was used to create a mathematical model.

Paci et al. found that the larger the cargo, the more nuclear localization signals it needed to be efficiently transported across the membrane into the nucleus. This is because inserting big cargoes into the NPC requires more energy. Therefore, by increasing the number of surface signals transporter proteins can bind to, larger molecules are able to interact with the NPC and generate the energy required for crossing.

These findings improve our current understanding of how nuclear transport could be hijacked by viruses. It could also help scientists who are developing targeted nanoparticles to deliver therapies for genetic conditions to the nucleus.

recent improvements in electron tomography (ET), paired with X-ray crystallography, have greatly expanded our knowledge on the organisation of these folded components of the NPC (*von Appen et al., 2015*; *Szymborska et al., 2013*; *Lin et al., 2016*; *Kosinski et al., 2016*). This pore-like scaffold is filled with multiple copies of ≈10 different intrinsically disordered proteins, known as FG nucleoporins (FG Nups), which form the NPC permeability barrier. FG repeats have been estimated to be at concentrations in the mM range inside the NPC (*Aramburu and Lemke, 2017*; *Frey and Görlich, 2007*). Our structural knowledge about the actual transport conduit compared to the scaffold is much lower, as its dynamic nature leads to a loss of electron density in the averaging process inherent to ET, leaving a ≈40 nm wide 'hole' inside the structural map of the NPC tomogram. As the transport of many large cargoes is believed not to irreversibly alter the structure of the NPC, substantial amounts of FG Nups mass must be displaced in order to facilitate such transport events. In addition to dynamics in the permeability barrier, dilation mechanisms in the scaffold structure itself have also been suggested (*Beck and Hurt, 2017*).

Despite its high biological relevance, nuclear transport of large cargoes is still poorly understood. In order to address this gap, we designed a set of large model cargoes based on capsid-derived structures. In contrast to using fully physiological large cargoes, such as complete viruses, this strategy enabled us to titrate key features such as size, number of binding sites and surface properties. This reductionist approach opened the possibility to experimentally measure a rigorous set of biophysical parameters. We used a combination of spectroscopy and semi-automated microscopy assays to investigate the kinetics of nuclear import of cargoes ranging from 17 to 36 nm in diameter and with a number of NLSs between 0 and 240 in permeabilised cells. Our results uncovered the quantitative dependence of cargo size and NLS number in an understudied size range. The results are rationalized using a minimal physical model of nuclear transport that takes into account the

energy gain from NTR binding to FG motifs, and the free energy cost needed for the insertion of a large particle into a densely filled channel.

## Results and discussion

### A large cargo toolkit for nuclear transport studies

We first aimed to develop a set of model import cargoes with known size and tuneable number of NLSs (#NLSs) on their surface. Naturally occurring cargoes with multiple NLSs, such as proteasomes, pre-ribosomes, mRNA or RNA-protein complexes do not offer the possibility to control both properties reliably at the same time. Vice versa, for artificial large substrates, like quantum dots or gold nanoparticles, it can be challenging to tune size and #NLSs and extensive functionalisation is typically required. Thus we turned to viral capsids, which are known to self-assemble from one or few proteins into large structures of fixed size. We screened the literature for capsid-like particles obeying the following criteria: i) Large-scale high yielding recombinant expression is possible in an expression host like *Escherichia coli*. ii) Surface modification via a unique residue is possible. Thus, we focused on systems with existing crystal and/or EM structures and checked for single functional surface exposed cysteines or the possibility of mutating another residue to one with no impact on capsid assembly. iii) Capsid is stable at physiological conditions. iv) Capsid diameter is between 15 nm and 40 nm: this size range focuses on rather uncharted territory, with its upper limit reported to be the largest size of cargoes transported by the NPC (Hepatitis B virus, *Panté and Kann, 2002*). As a result, the following four icosahedral shaped capsids of different size were selected for this study (*Figure 1*).

MS2$^{S37P}$ (diameter 17 nm): This capsid is derived from the bacteriophage MS2, formed by a single coat protein with a point mutation S37P. The coat protein assembles into dimers and then into 12 pentamers yielding an icosahedron with a total of 60 copies (*Asensio et al., 2016*). A cysteine mutation (T15C) that had previously been shown not to interfere with capsid assembly was introduced to allow surface tagging via maleimide labelling (*Peabody, 2003*).

I53-47 (diameter 23 nm): This artificial capsid is derived from de novo designed capsids developed by the Baker lab (*Bale et al., 2016*). The I53-47 variant is formed by two different proteins (*chain A* and *chain B*), occurring in 60 copies each and organised into 12 pentamers and 20 trimers. A cysteine mutation exposed on the capsid surface was introduced in *chain B* (D43C), following the recent work where different surface mutations were introduced in a similar capsid variant (*Butterfield et al., 2017*). We note that another synthetic capsid of a similar type but 27 nm in diameter, I53-50, could not be specifically labelled and thus was not included in this work (*Figure 1—figure supplement 1*).

MS2 (diameter 27 nm): This capsid is derived from the wild-type bacteriophage MS2 coat protein, which in total of 180 copies assembles into dimers and then into an icosahedron with 12 pentameric and 20 hexameric faces. The same cysteine mutation as in MS2$^{S37P}$ (T15C) enabled tagging via maleimide labelling (*Peabody, 2003*).

Hepatitis B capsid (diameter 36 or 32 nm depending on isoform): This capsid is based on an assembly-competent truncated version of the HBV core protein (aa 1–149). This truncation leads to higher levels of bacterial expression and to a predominance of the T = 4 capsid (36 nm) with no obvious change in capsid morphology (*Zlotnick et al., 1996*). The core protein thus assembles mainly into 12 pentameric and 30 hexameric units, for a total of 240 copies. The truncation also removes the C-terminal native NLS (which can be buried inside the capsid), enabling a complete control over the number of exposed NLSs via surface engineering. A cysteine mutation (S81C) was introduced into an exposed loop of the core protein (c/e1 epitope) to allow surface tagging via maleimide labelling. The Hepatitis B capsid is frequently quoted as the largest cargo known to pass the NPC intact (*Panté and Kann, 2002*), and constitutes the upper limit of the cargoes we investigated.

After successful purification, the next step was to engineer the capsid surface with a fluorescent dye and with NLSs. As detailed in the methods, the use of tangential flow for sample concentration and buffer exchange turned out to be of highest practical relevance to purify preparative amounts of intact capsids for further labelling reactions. We chose maleimide reactive dyes and a synthetic maleimide reactive NLS, with a sequence known to bind tightly to Importinα, which binds to Importinβ via its IBB domain (*Hodel et al., 2001*). Capsids were labelled with suitable mixtures of dye and NLS peptide simultaneously.

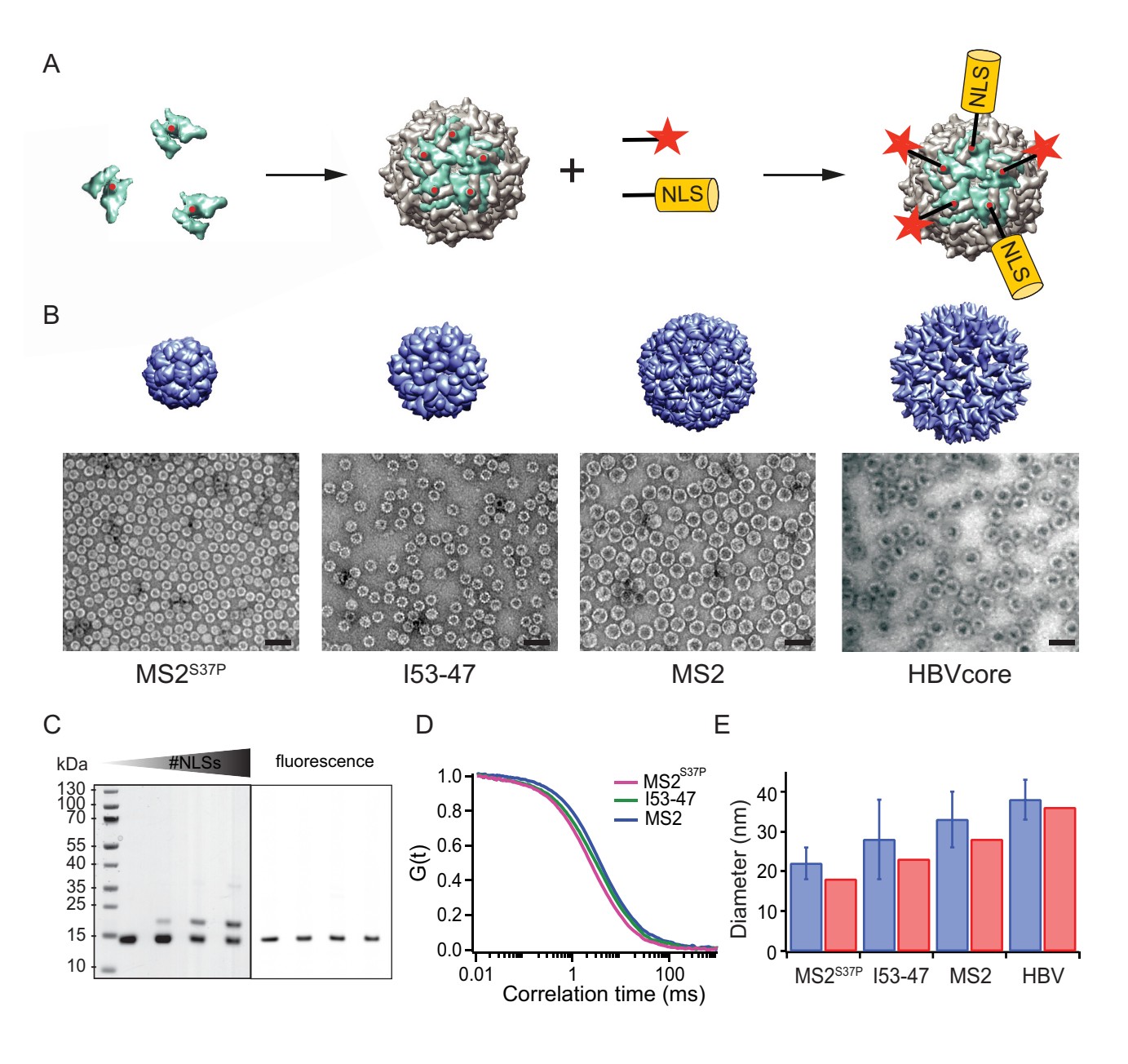

**Figure 1.** A large cargo 'toolkit' for nuclear import studies. (**A**) Schematic representation of the mixed labelling reaction with maleimide reactive NLS peptide and maleimide reactive fluorescent dye. The capsid protein, containing a cysteine mutation (in red), self-assembles into a capsid. The purified capsids are then labelled with a mixture of dye and NLS peptide, in different ratios according to the desired reaction outcome. (**B**) Capsid structures rendered in Chimera (*Pettersen et al., 2004*) (top) and EM images of the purified capsids (bottom). The scale bar corresponds to 50 nm. (**C**) SDS-PAGE gel of MS2[S37P] samples with increasing number of NLS peptides attached (top band). The lower band corresponds to a capsid protein tagged with dye or no dye, but 0 NLS. The upper band corresponds always to the capsid protein without any dye, but NLS, as evident from the fluorescent scan on the right side. (**D**) Representative FCS autocorrelation curves for the MS2[S37P], I53-47 and MS2 capsids. The curves were fitted with a diffusion model to calculate the capsid brightness and concentration. (**E**) DLS quantification of capsid diameters (blue bars) compared with reference values from literature and structural information (red bars).

The online version of this article includes the following figure supplement(s) for figure 1:

**Figure supplement 1.** I53-50 capsid.

*Figure 1* summarises the labelling scheme used for all capsids and its characterisation. *Figure 1B* shows negative staining EM images of capsids after purification and labelling, visualizing intact capsids with the expected diameter. To guarantee the robustness of the quantitative experiments, it was crucial to determine each capsids' fluorescence brightness (i.e. how many dyes are attached to one capsid) as well as the #NLSs. The #dyes/capsid was determined via fluorescence correlation spectroscopy (FCS), a widely employed biophysical tool to probe brightness and concentration of a freely diffusing species (*Figure 1D*, *Table 1*). FCS can also be used to estimate the size and size distribution (such as substantial contaminations of other species than intact capsids) of the samples, which was found to be in line with the high purity indicated by the EM micrographs. Additional DLS (dynamic light scattering) studies were employed to further validate capsid diameter in solution and presence of intact capsids as the dominant species (*Figure 1E*). The #NLSs was determined from gel shift assays, as NLS-labelled capsid monomers migrate substantially different than their unlabelled counterparts. In contrast, the dye labelling did not alter capsid monomers mobility on gel (*Figure 1C*). Estimated #NLSs and #dyes/capsid are listed for all samples in *Table 1*. The presence of a single cysteine per monomer ensures that each is labelled either with NLS or dye, but not both: in this way, unassembled monomers cannot be fluorescently detectable NLS-dependent import

**Table 1.** Sample properties and parameters from fits of import kinetics.

Here we list all capsid sample properties (estimated #NLSs and #dyes per capsids), as well as all parameters extracted from fitting the import traces with an inverse exponential $I(t) = A + I_{MAX}(1 - e^{-\tau * t})$. The initial flux is calculated as $J = I_{MAX} * \tau$. When different biological replicates were measured for the same sample, the values indicate the average.

| | sample | #dyes | #NLSs | A | $I_{MAX}$ | $\tau$ | J | $\Delta G$ |
|---|---|---|---|---|---|---|---|---|
| MS2$^{S37P}$ | 1 | 23 | 0 | 0.24 | 0.39 | 0.022 | 0.01 | 6.34 |
| | 2 | 15 | 14 | 0.65 | 0.94 | 0.054 | 0.05 | 4.54 |
| | 3 | 30 | 19 | 1.12 | 12.44 | 0.041 | 0.49 | 2.01 |
| | 4 | 34 | 23 | 0.64 | 16.63 | 0.053 | 0.88 | 1.25 |
| | 5 | 25 | 29 | 0.33 | 28.91 | 0.022 | 0.64 | 1.51 |
| | 6 | 38 | 38 | 1.45 | 43.59 | 0.037 | 1.60 | −0.02 |
| | 7 | 10 | 54 | 2.33 | 49.76 | 0.029 | 1.37 | 0.32 |
| I53-47 | 8 | 24 | 0 | 0.18 | 1.58 | 0.018 | 0.03 | 5.27 |
| | 9 | 30 | 15 | 0.80 | 3.71 | 0.085 | 0.32 | 3.19 |
| | 10 | 31 | 18 | 3.81 | 3.89 | 0.056 | 0.22 | 3.29 |
| | 11 | 36 | 22 | 2.34 | 2.95 | 0.080 | 0.23 | 3.36 |
| | 12 | 16 | 22 | 2.01 | 5.35 | 0.060 | 0.32 | 3.04 |
| | 13 | 31 | 25 | 1.83 | 2.91 | 0.094 | 0.27 | 3.15 |
| | 14 | 6 | 30 | 2.27 | 7.22 | 0.063 | 0.45 | 2.41 |
| | 15 | 3 | 35 | 1.18 | 16.82 | 0.048 | 0.81 | 1.49 |
| | 16 | 8 | 37 | 1.21 | 6.67 | 0.053 | 0.35 | 2.28 |
| | 17 | 3 | 37 | 2.16 | 13.80 | 0.059 | 0.81 | 1.52 |
| | 18 | 10 | 41 | 1.13 | 13.92 | 0.057 | 0.79 | 1.54 |
| | 19 | 8 | 44 | 0.31 | 11.78 | 0.038 | 0.45 | 2.00 |
| MS2 | 20 | 50 | 0 | 0.07 | 0.18 | 0.038 | 0.01 | 6.88 |
| | 21 | 38 | 42 | 0.47 | 0.52 | 0.106 | 0.05 | 4.94 |
| | 22 | 58 | 54 | 0.19 | 1.07 | 0.241 | 0.26 | 2.95 |
| | 23 | 44 | 57 | 0.07 | 1.83 | 0.074 | 0.13 | 3.49 |
| | 24 | 52 | 77 | 0.47 | 1.31 | 0.072 | 0.09 | 3.96 |
| | 25 | 61 | 86 | 0.55 | 2.67 | 0.042 | 0.11 | 3.72 |
| | 26 | 57 | 93 | 0.38 | 2.04 | 0.051 | 0.10 | 3.88 |
| | 27 | 54 | 98 | 0.27 | 3.82 | 0.033 | 0.12 | 3.49 |

substrates. We note that labelling with a synthetic NLS pre-tagged with a dye was found to be impractical in preliminary experiments, as the unreacted species can contribute to elevated background fluorescence in the nucleus.

## Import kinetics of large cargoes are tuned by size and NLS numbers (#NLS)

The different labelled capsids samples (total 30, *Table 1*) were subjected to nuclear import assays using the widely employed permeabilised cell assay (*Adam et al., 1990*). *Figure 2A* outlines the details of the experiments. In brief, mild digitonin treatment was used to permeabilise the plasma membrane of HeLa cells, leaving the nuclear envelope intact. In these conditions, functional nucleo-cytoplasmic transport can be reconstituted for a few hours by adding the key components of the transport machinery: Importinβ, Importinα, RanGDP, NTF2 (a NTR which allows recycling of RanGDP) and GTP to the cells. Intactness of the nuclear envelope and functional nuclear transport were always validated by a set of control experiments using fluorescently labelled dextran and model cargoes (see Materials and methods). As shown in *Figure 2B* exemplarily for the MS2$^{S37P}$ and MS2 capsids, cargoes labelled with NLSs showed an increased nuclear accumulation over time, indicative of functional nuclear import.

Experiments were performed on a semi-automated confocal microscope, recording time-lapse images over several cells and different field of views (error bars correspond to standard deviations between different FOV). Note that for practical reasons, imaging always started ~ 2 min after addition of the transport mix to the cells. This timing offset was accounted for by an offset fitting parameter $A$ in our fit equation ($I(t) = A + I_{MAX}(1 - e^{-\tau * t})$).

Besides the nuclear signal, we also recorded the nuclear envelope and cytoplasmic signals using suitable imaging masks (*Figure 2C*, Materials and methods and *Source code 1* for details). We took precautions to distinguish nuclear fluorescence from nuclear envelope fluorescence by eroding the nuclear mask to a region furthest away from the rim. This turned out to be important, as some capsids showed nuclear envelope targeting but no substantial accumulation into the nucleoplasm (for instance, HBV and MS2 capsids with few NLSs). In addition, this method enabled us to discriminate nuclear signal from sticking of capsids to the cytoplasm, which was observed in some cases.

*Figure 2D* summarises the three kinetic traces that were obtained from a typical experiment. In the representative experiment shown for a MS2$^{S37P}$ capsid sample, the cytoplasmic fluorescence stayed constant, while nuclear envelope signal increased pointing to recruitment and accumulation of capsids at the NPCs. The red curve shows the import kinetics of capsids into the nucleus. *Figure 2—figure supplement 1* shows additional control experiments (addition of the Importinα export receptor CAS to transport mix and excess of GTP or Importinα) to establish that the observed saturating nuclear import depends on the substrate size and #NLSs and is not due to any of the components in the transport mix becoming limiting during the course of the experiment.

To further support our findings under fully physiological conditions, we carried out microinjection of representative capsid samples in starfish oocytes to observe their nuclear accumulation in live cells. The results of these experiments are presented in *Figure 2—figure supplement 2* and are qualitatively in agreement with the quantitative nuclear import assays in permeabilised cells described in the next paragraph.

*Figure 3* shows representative nuclear import data for the three kinetically investigated capsids MS2$^{S37P}$, I53-47 and MS2 (see *Figure 3—figure supplement 1* for full dataset). The results for HBV capsids will be discussed in the next paragraph. *Figure 3* panel A displays typical confocal images of cargoes with different #NLSs and panel B shows representative nuclear kinetic traces extracted from semi-automated microscopy. *Figure 3—figure supplement 1* shows the full dataset overlaid with the mono-exponential fits. In absence of NLSs (0 NLSs), all capsids localised to the cytoplasm and no targeting to the nuclear envelope or accumulation in the nucleus was observed, in line with an Importin-dependent pathway. With increasing #NLSs present on the capsid surface we observed progressive nuclear envelope targeting, and eventually, efficient accumulation of cargo in the nucleoplasm. Strikingly, the #NLSs required to observe similar behavior with different capsids scaled dramatically with cargo size, as can be seen by comparing for example the I53-47 sample image with 35 NLSs and the MS2 one with 86 NLSs. The observation of robust bulk import for all capsid constructs with sufficiently high #NLSs highlights another benefit of using viral capsids as large cargo models:

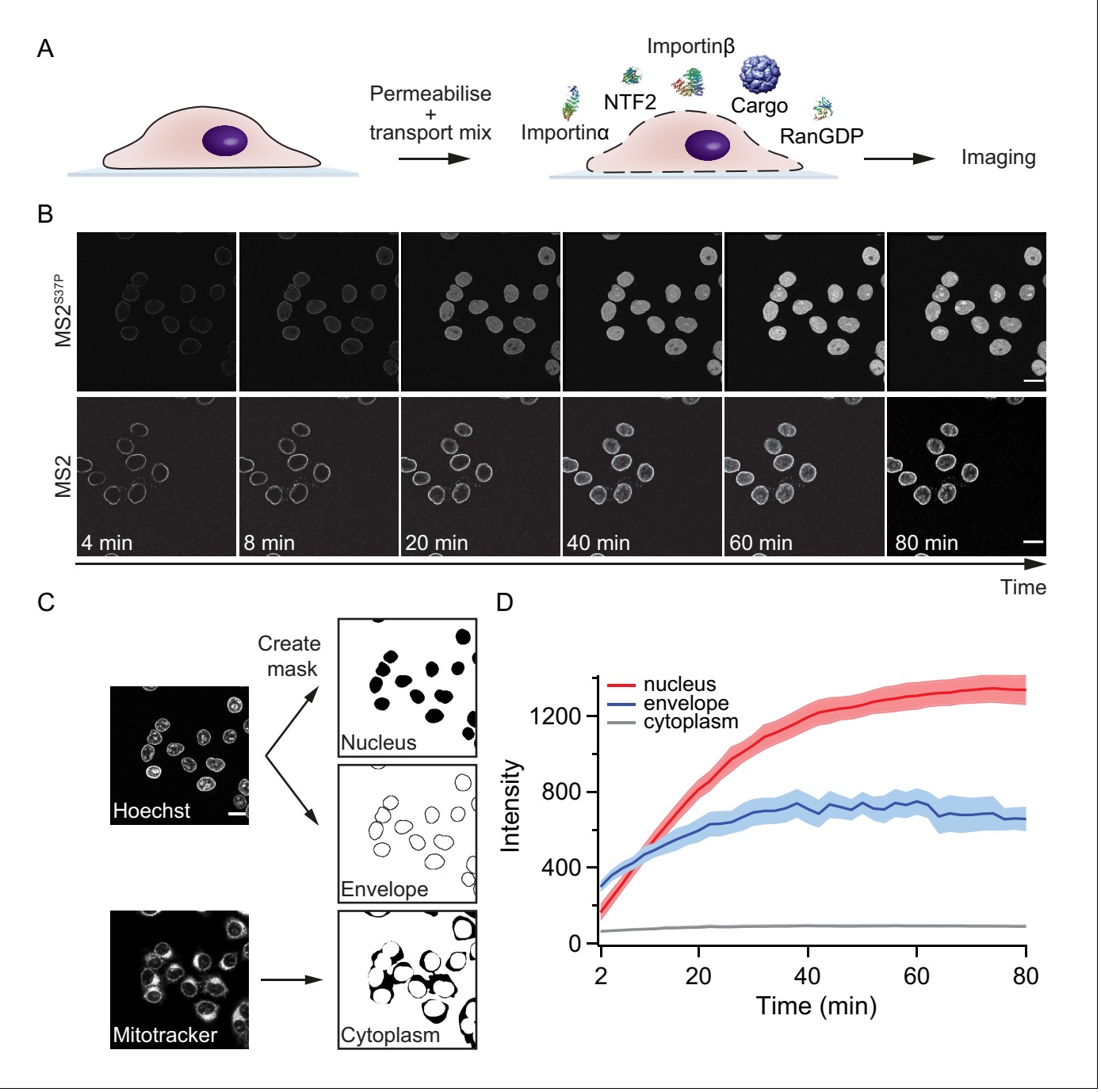

**Figure 2.** Pipeline for import kinetic experiments. (A) Scheme of the transport assay experiment: HeLa cells were permeabilised and incubated with a transport mix containing the cargo of interest, nuclear transport receptors and energy. Confocal images were acquired in 12 different areas every 2 min, for 80 min in total. (B) Representative time-lapse snapshots of cargo import (MS2$^{S37P}$ and MS2 capsids). The scale bar corresponds to 20 μm. (C) Overview of the image analysis pipeline for import kinetics experiments. Two reference stain images (Hoechst and MitoTracker) were segmented and used to generate three masks corresponding to the regions of interest: nucleus, nuclear envelope and cytoplasm. The masks were then applied to the cargo images to calculate the average intensity in the different regions. (D) Representative raw import kinetics traces for the three cellular compartments of interest. Note that imaging starts after 2 min of adding the transport mix to the cells. Curves depict the average fluorescence measured in the different regions; the shaded areas represent the standard deviation over 12 areas.

The online version of this article includes the following figure supplement(s) for figure 2:

**Figure supplement 1.** Control experiments in permeabilised cells.

**Figure supplement 2.** Microinjection of capsids in live starfish oocytes.

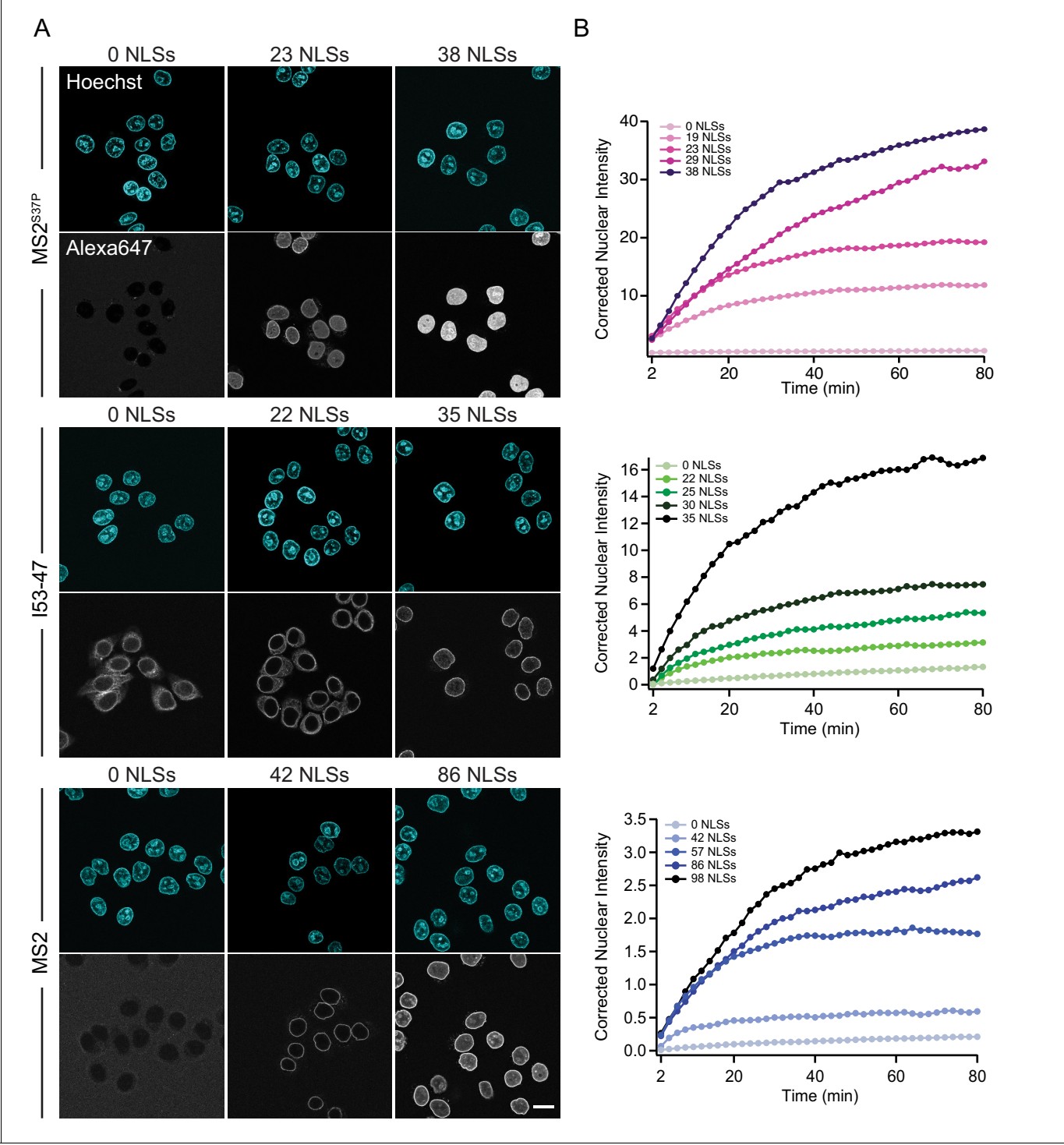

**Figure 3.** The import kinetics of large cargoes is tuned by the NLS number. (**A**) Confocal images of nuclear import of the different large cargoes. Cells were incubated for up to 1.5 hr with capsids tagged with different number of NLS peptides on their surface. All cargoes displayed a distinct NLS-dependent behaviour. The scale bar corresponds to 20 μm. (**B**) Representative nuclear import traces for the three large cargoes labelled with increasing amount of NLS peptides. The corrected nuclear intensities are obtained by background-subtracting the raw nuclear intensities, scaling them according to capsid brightness (#dyes) estimated from FCS (**Table 1**) and subtracting the initial offset $A$ determined by the mono-exponential fit, to better compare the import efficiencies. The corrected intensities are proportional to capsid concentration and allow us to compare the import efficiency of the different samples. See **Figure 3—figure supplement 1** for the full dataset displayed without offsetting by $A$ and overlaid with mono-exponential fits. The online version of this article includes the following figure supplement(s) for figure 3:

*Figure 3 continued on next page*

*Figure 3 continued*

**Figure supplement 1.** Entire import kinetic dataset.

in a previous study using coated quantum dots (18 nm) no bulk import could be detected but only rare import events were captured by advanced single molecule technologies (*Lowe et al., 2010*).

## Modified HBV capsids are targeted to NPCs but do not accumulate into the nucleoplasm

We next used the established pipeline to investigate the transport of HBV capsids, achieving a maximum of 50% capsid monomer labelling (120 NLSs). The capsids were targeted to the nuclear envelope; however, no bulk nuclear import could be detected (*Figure 4*, first row). As we were not able to further increase the #NLS with our chemical labelling strategies, and we wondered whether 120 NLSs might still be insufficient, we resorted to genetic tools to achieve the full coverage of 240 NLSs per capsid. To do this, we designed a capsid based on the SplitCore construct (*Walker et al., 2011*), in which a core-GFP fusion protein was split into two halves that self-assemble before forming the capsid. This exposes a free C terminus, which we exploited to introduce an NLS. Also for this

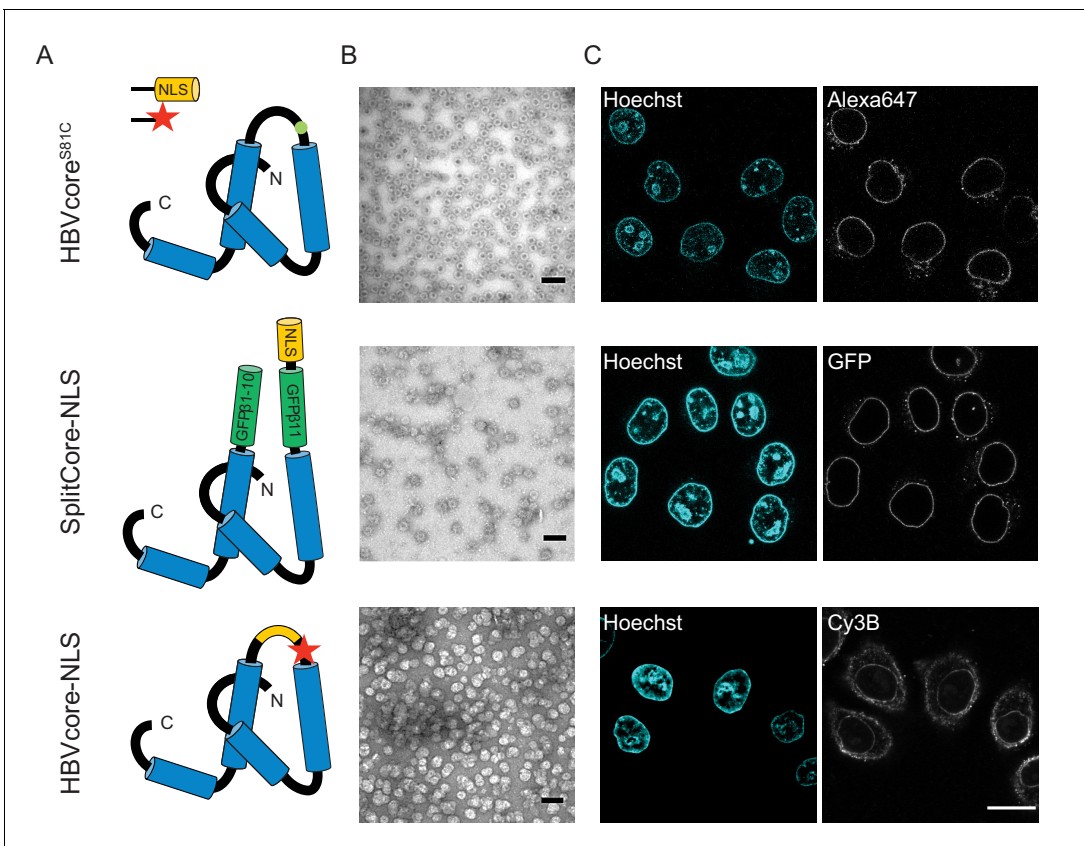

**Figure 4.** The NLS-engineered Hepatitis B capsid is not imported in the nucleus of permeabilised cells. Following the same labelling approach as described in *Figure 1*, HBV capsids with up to 120 NLSs were generated (first row). In order to test capsids with a higher number of NLSs exposed on the surface, we designed two additional versions of the HBV core protein with a direct NLS insertion (total of 240 NLSs). The middle row shows a construct based on the SplitCore-SplitGFP (*Walker et al., 2011*), where the HBV core protein is split via artificial stop and start codons into two halves and fused to a split-GFP (GFPβ1–10 and GFPβ11), to which we further added an NLS. Once co-expressed, the two core-GFP halves self-assemble into capsid-like particles. The last row shows a construct where the NLS is inserted in the c/e1 epitope loop of the core protein (orange loop) and a cysteine mutation is introduced to perform labelling with a dye (red star). All capsids were targeted to the nuclear envelope but did not give rise to bulk nuclear accumulation in import experiments using permeabilised cells. (**A**) Schematic representations of the different HBV core protein constructs. (**B**) EM images of the purified capsids. The scale bar corresponds to 100 nm. (**C**) Confocal images of capsid import experiments after 1.5 hr. The scale bar corresponds to 20 µm.

capsid, we did not observe any bulk import. However, the slightly increased size due to the GFP could potentially push this capsid over the maximum NPC transport size limit. We thus tested another strategy, and introduced an NLS into an exposed capsid loop (*Figure 4*, last row). Again, no functional bulk import could be observed. EM showed that the engineered capsids are less homogenous, but still a large number of intact capsid was observed. Hence we conclude that none of the tested HBV capsids constructs can functionally be enriched in the nucleus. As the chances that our careful modifications rendered the HBV capsid transport-incompetent seem rather low, our data is in line with studies that suggest that only the mature infectious virus can translocate through the NPC into the nucleoplasm (*Rabe et al., 2003*; *Kann et al., 1999*). Our results are consistent with EM data of intact HBV capsids entering the NPC barrier, (*Panté and Kann, 2002*) as we also see strong NE accumulation. However, additional mechanisms would be required for cargo release into the nucleoplasm such as the previously reported structural destabilisation of mature capsids (*Cui et al., 2013*) or other mechanism that can disassemble capsid that are docked at the NPC. Collectively, this suggests that 36 nm capsids might be able to enter the NPC barrier, but are too large to pass the NPC intact into the nucleus (i.e. undock or release). We, thus, focus our global quantitative analysis on the three capsids for which we could experimentally identify conditions of functional import and nuclear enrichment.

## Quantitative analysis of nuclear import in relation to cargo size and #NLSs

Our results on large cargo import kinetics (*Figure 3*) highlight the strikingly different #NLS requirements for the nuclear import of differently sized cargoes. We formulate here a biophysical model that considers the translocation of a large 'spherical object' through the crowded NPC permeability barrier (scheme in *Figure 5A*) and enables us to extract key information about the energetics of transport from our kinetic data.

The final steady state accumulation and the late kinetics of the capsid import are affected by a number of factors that are still incompletely understood – such as the competition between Ran and NTRs for the cargo, the back leakage of the cargo into the cytoplasm and potential clogging of the pores by the capsids (*Kim and Elbaum, 2013a*; *Kim and Elbaum, 2013b*). For this reason, we focus our quantitative analysis on the initial flux $J$ (slope of the kinetic curve at the initial time point). Unlike the steady state accumulation, the initial flux $J$ of cargoes into the nucleus is independent of the rates of cargo-NTR dissociation kinetics and is less affected by any potential rate-limiting steps in the Ran cycle (*Kim and Elbaum, 2013a*; *Kim and Elbaum, 2013b*; *Görlich et al., 2003*). To this end, all nuclear import curves were fitted with a mono-exponential kinetic model $I(t) = A + I_{MAX}(1 - e^{-\tau * t})$, with $I_{MAX}$ being the plateau value reached by the fit at infinity, $\tau$ the reaction constant with units 1/s and $A$ is the offset parameter. $A$ accounts for any nonzero offset, which could be due to: i) initial recruitment of the cargoes to the cells and nuclear envelope. ii) limiting accuracy in pipetting and sample mixing (there is a 2-min delay in our experiments between the addition of the sample and the start of imaging) and for slightly different background levels due to non-specific adhesion of some samples to cellular structures. $A$ is thus fitted in every experiment and not expected to be a constant. The initial flux can be calculated from the fit parameters as $J = I_{MAX} * \tau$ (see *Table 1* for values of all fit parameters). We emphasize that the mono-exponential fit is a mathematical tool to estimate the initial flux from the data. Calculating the initial flux from the mono-exponential fits was more robust than the alternative of measuring the initial flux directly from a linear fit of the first few data points, since the timing resolution of the experiment and the accuracy of defining the zero time point when mixing the cargo with the cells was limited. We note that more complex fits, such as bi-exponential fits have been discussed in the literature to include additional effects such as cargo leaking back into the cytoplasm. (*Kim and Elbaum, 2013a*; *Kim and Elbaum, 2013b*). In *Supplementary file 1*, we further compare bi-exponential and the mono-exponential fits. The initial rates for all samples are plotted in *Figure 5C* (experimental data displayed as dots). We also note that despite the samples having different labelling ratios (see #dyes, *Table 1*), we confirmed that there were no global correlations between overall #dyes/capsid ratio and extrapolated rate ($R^2$=0.14).

Based on extensive previous theoretical and experimental work on the NPC (*Iyer-Biswas and Zilman, 2016*; *Zilman, 2009*; *Zilman et al., 2007*; *Berezhkovskii et al., 2002*; *Pagliara et al., 2013*),

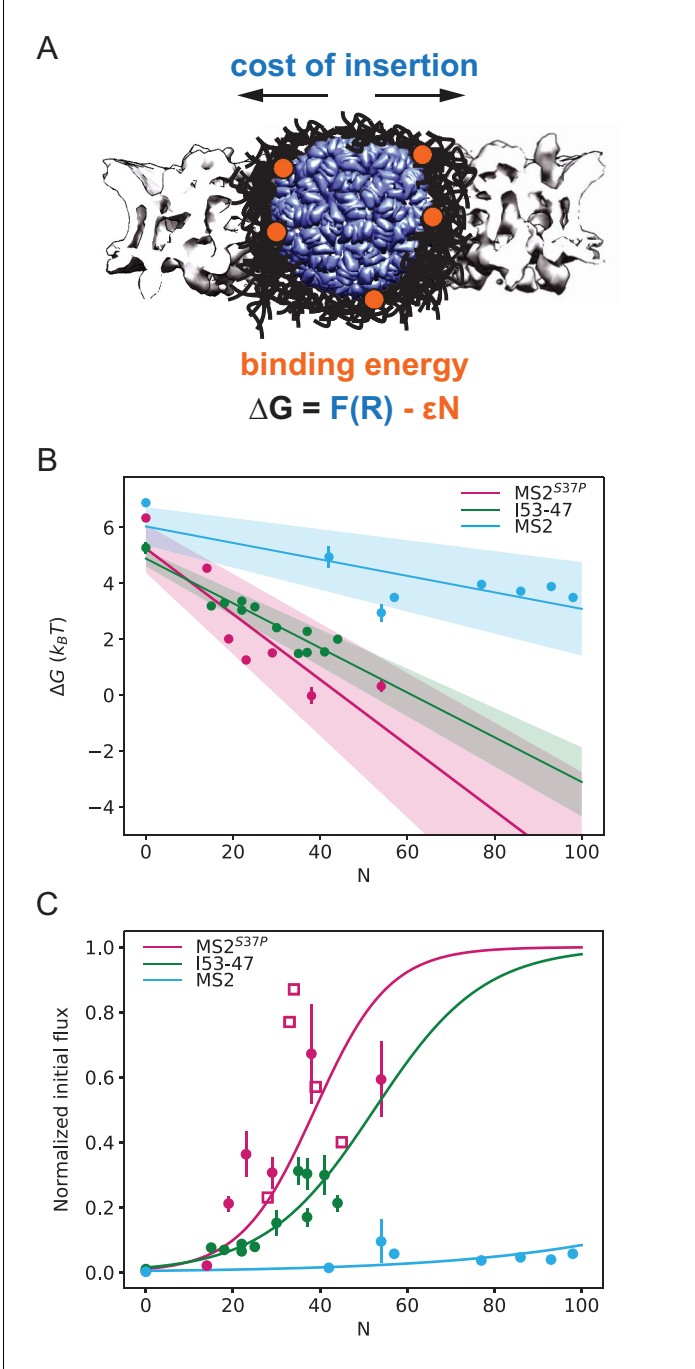

**Figure 5.** Effect of cargo size and number of NLSs (#NLSs) on import kinetics and biophysical model. (**A**) Cartoon of the determinants for large cargo import: the free energy cost of inserting a large cargo into the dense FG Nup barrier must be compensated by the binding to FG Nups via multiple NTRs (binding sites represented in orange, NTRs omitted for simplicity). The NPC scaffold structure is from EMD-8087. (**B**) Dependence of $\Delta G$ on the capsid size and #NLS for $a_{Ran} = 2$. Shaded regions show one standard deviation of $F(R)$ and $\epsilon$. Fitted values for $F(R)$ and $\epsilon$ are shown in **Table 2**. (**C**) Initial flux (corresponding to the slope of the kinetic curve at the initial time point) modelled as $J \propto \frac{1}{a_{Ran} + e^{F(R) - \epsilon N}}$ overlaid on the (normalised) experimental data (dots). Additional experiments with $MS2^{S37P}$ capsids containing additional charges are overlaid and shown as squares. Whenever independent biological replicates were available, the initial flux is calculated as an average and shown with the error extracted from the technical replicates (12 areas imaged in each experiment). In **Figure 5—figure supplement 5** we show that the uncertainty between different cells imaged in a single experiment captures well the variability of independent experiments.

*Figure 5 continued on next page*

*Figure 5 continued*

The online version of this article includes the following figure supplement(s) for figure 5:

**Figure supplement 1.** Results from biophysical model with $a_{Ran} = 1$.
**Figure supplement 2.** Results from biophysical model where the data point for #NLS=0 is excluded from the fit.
**Figure supplement 3.** Non-uniform distribution of FG Nups along the pore: theoretical model.
**Figure supplement 4.** Comparison of large cargo surface properties.
**Figure supplement 5.** Comparison of biological and technical replicates.

the initial flux $J$ can be approximated as $J = \frac{k_{ON}c}{a_{Ran} + e^{\Delta G}}$, where $k_{ON}$ is the rate of cargoes reaching the NPC entrance, $c$ is the concentration of cargoes in the cytoplasm, $a_{Ran}$ is a number between 1 and 2 depending on the availability of RanGTP at the nuclear exit ($a_{Ran} = 2$ corresponds to the absence of RanGTP, and $a_{Ran} = 1$ corresponds to RanGTP always being immediately available at the nuclear exit). $\Delta G$ is the effective average non-equilibrium free energy potential of the cargo inside the NPC (expressed in units of $k_B T \approx 0.6\ kcal/mol$; a conceptually similar expression was used in *Frey and Görlich, 2007* to analyse the transport of cargoes through FG Nup 'hydrogels'. This expression mathematically describes the fact that the probability of a particle that impinges on the NPC entrance to actually translocate to the other side is $P_{tr} = \frac{1}{a_{Ran} + e^{\Delta G}}$ due to the random nature of the diffusive motion inside the NPC. For cargoes that are strongly repelled by the FG Nup network, $\Delta G \gg 1$, and the flux is exponentially inhibited, as $J \sim e^{-\Delta G}$. By contrast, for cargoes that interact attractively with the FG Nups, $\Delta G < 0$ resulting in significant flux through the pore. However, the flux can be significant and well detectable already for $\Delta G \simeq 1 k_B T$. This expression remains valid for the low concentrations studied here and the intermediate values of $\Delta G$ appropriate for our capsids. For higher concentrations or higher #NLS, which we did not experimentally assess, additional corrections may need to be introduced (*Zilman et al., 2007*; *Pagliara et al., 2013*).

From experimental and theoretical studies (*Gu et al., 2017*; *Ghavami et al., 2016*; *Vovk et al., 2016*; *Maguire et al., 2020*), in the first approximation the main components of $\Delta G$ are: 1) the cost of insertion of the capsid into the FG assembly $F(R)$, arising from the entropic cost of FG Nup displacement, osmotic pressure and the effective surface tension penalty, and 2) the effective energetic/enthalpic gain $\epsilon$ from the attractive contacts formed between the NTR binding sites and FG repeats, which can partially compensate for the cost of insertion. $\Delta G$ can also include non-equilibrium logarithmic corrections arising from the dependence of the diffusion coefficient on the cargo size and the #NLSs (*Zilman, 2009*; *Maguire et al., 2020*).

Previous studies indicate that the cost of insertion, $F(R)$, increases with the particle size (*Gu et al., 2017*; *Ghavami et al., 2016*; *Vovk et al., 2016*). In simple situations such as the partitioning of a relatively small spherical particle into a polymer brush or a polymer-coated channel representing the NPC, this cost scales as $F(R) \propto R^\alpha$ with $1 \leq \alpha \leq 3$, where $R$ is the radius of the particle (*Gu et al., 2017*; *Ghavami et al., 2016*) but the exact form of the dependence on $R$ is unknown for very large cargoes studied here. The energetic gain is given by the total average energy of binding between NTRs and FG repeats, which, in the first approximation, for independent binding is expected to be proportional to the #NLSs on the particle surface, which we denote here as $N$. Combining these terms yields $G = F(R) - \epsilon N$, where $\epsilon$ is the effective binding interaction energy of an NTR with the FG environment; $\epsilon$ is proportional to the product of the density of the FG motifs in the pore $\phi$, and $\epsilon_0$, the bare average interaction energy of an NTR with an FG motif, so that $\epsilon = \epsilon_0 \phi$. When the insertion cost term dominates, $\Delta G$ is large and positive, and the initial accumulation rate is low. On the other hand, when the energetic gain term dominates (large $N$), $\Delta G$ decreases and eventually becomes negative, and the initial accumulation rate is high.

In order to gain insight into the transport mechanism, we analysed the experimental data using the minimal model described above. First, we inverted the equation for the initial flux to obtain the $\Delta G$ values as a function of the #NLSs ($N$) for the three capsids of different sizes. For each capsid size, $\Delta G$ values were fit with a straight line, obtaining the values for $F(R)$ and $\epsilon$ from the $y$-intercepts and slopes, respectively (*Figure 5B*). We assumed here $a_{Ran} = 2$; results for $a_{Ran} = 1$ are very similar, as shown in *Figure 5—figure supplement 1*. The actual value of the initial flux depends on the $k_{ON}$ (see above) and the number of the NPCs in the nuclear envelope – variables that are hard to estimate experimentally. Thus, for the purpose of comparison with the model, the data were normalised

to the maximal observed value among all technical replicates that was still within 95% confidence interval for that value. The conclusions of the analysis were robust with respect to the choice of the normalisation constant.

*Figure 5C* shows the experimentally measured initial flux $J$ data (dots) overlaid with the theoretical equation for $J$ using the values of $F(R)$ and $\epsilon$ obtained from the fit (*Figure 5B*, parameters values are listed in *Table 2*). The fits in both figures agree well with our experimental data. Consistent with the theoretical expectations, the cost of insertion $F(R)$ was the highest for the largest capsid. The differences between the insertion costs for the two smaller capsids were within the error bars. To control for the possibility that the similar values of $F(R)$ observed for all three capsids are an artefact of the limitations on the experimental accuracy at very low fluxes, we repeated the model fit, excluding the #NLS=0 point, which resulted in essentially the same fitting parameter values (*Figure 5—figure supplement 2*). Another possibility is that for such large capsids the insertion cost saturates to a plateau value at maximal FG Nup compression.

Surprisingly, the $\epsilon$ values were different for different capsids, with the $\epsilon$ for the MS2 (largest) capsid substantially lower than those for MS2[S37P] and I53-47 capsids. At first glance, one would expect the main difference in the fluxes of capsids of different size to stem from the difference in the insertion cost $F(R)$, while the interaction energy would be relatively unaffected by the particle size. It was also surprising that significant accumulation in or near the nuclear envelope was observed even for the cargo samples whose interaction with the NPC is insufficient to cause substantial nuclear accumulation (*Figure 2D*).

To further understand the implications of these findings, we extended the model to include a variation in the FG Nup density along the pore. Our model is a variant of previously postulated 'vestibule'/'docking'models (*Tagliazucchi et al., 2013*; *Tu et al., 2013*; *Lowe et al., 2015*), with a central 'barrier' region with high density of FG Nups and correspondingly high insertion cost, and a 'vestibule' outside the NPC (corresponding to a low density cloud of FG Nups extending into the cytoplasm). The capsids weakly bind in the vestibule but experience no insertion cost as FG Nups and capsids are unconstrained by the NPC scaffold in this region. Bridging between the barrier and the vestibule there are narrow transition regions at the NPC peripheries, with a medium density of FG Nups and correspondingly low insertion cost. As shown in *Figure 5—figure supplement 3*, this simple extension of the model allows us to explain the fluxes of all capsids with the same value of the 'bare' NTR-FG binding energy $\epsilon_0$, as well as the accumulation in the cytoplasmic 'vestibule' even at low $\epsilon$. Assuming the average number of FG motifs in the pore ~3000 (*Tu et al., 2013*), corresponding to the average volume fraction/density $\phi = 0.01$, the obtained values of $\epsilon_0 \simeq 4 - 15\ k_BT$ are within the range of the common estimates of NTR-FG interaction strength (*Aramburu and Lemke, 2017*; *Tu et al., 2013*; *Kapinos et al., 2014*; *Eisele et al., 2010*; *Milles et al., 2015*). This analysis should be viewed with the caveat that this minimal model is likely to be modified in the future with more molecular details; we return to this point in the Discussion.

## Surface property effects on large viral import

Surface properties such as charge or hydrophobicity have been frequently indicated to influence the import properties through the nuclear pore complex of smaller cargoes, which in many cases were systematically assessed by creating large data sets in which the cargo properties were carefully studied and/or tuned (*Frey et al., 2018*; *Naim et al., 2009*; *Colwell et al., 2010*).

While our capsid study does not lend itself to similar high throughput screening of surface properties, we speculate on the role of surface properties for large cargoes based on a few observations and experiments. i) We found that our minimal physical model describes our experimental data well.

**Table 2.** Parameters from free energy fit.
Fitted values for $F(R)$ and $\epsilon$ values, for $a_{Ran} = 2$. The error corresponds to the standard deviation.

| Capsid | Diameter [nm] | F(R) [k$_B$T] | $\epsilon$ [k$_B$T] |
|---|---|---|---|
| MS2[S37P] | 17 | 5.2 ± 0.9 | 0.12 ± 0.03 |
| I53-47 | 23 | 4.9 ± 0.3 | 0.08 ± 0.01 |
| MS2 | 27 | 6.0 ± 0.7 | 0.03 ± 0.01 |

As the capsids all have a different and complex surface properties landscape (see *Figure 5—figure supplement 4*) this can be seen as an indicator that in the regime studied in this paper, the rules of large cargo transport might be dominated by the size of the capsid sphere and the number of NTRs that it can bind rather than direct interactions between the capsids and FG Nups due to surface effects. A potential exception could be at very low #NLS labelling regime, where the signal-to-noise ratio does not offer a detectable measurement of initial flux. ii) To substantially alter surface charges, we labelled capsids with a longer NLS peptide containing a linker with a negatively charged stretch of amino acids (DEDED). We focused on the MS2$^{S37P}$ capsid with high #NLS labelling, where consequently the largest number of additional charges could be included by this method. As shown in *Figure 5C* (charged capsid data shown as squares), we did not observe substantially different behaviours in capsids with and without the additional charges. We note that we faced practical hurdles in obtaining capsids with a positively charged linker due to precipitation/aggregation of the peptide during labelling and, thus, were not able to experimentally test this regime. iii) Simple geometrical considerations could also support that for large objects like our capsids the actual surface properties might be less relevant in the regime of large #NLS. If we just focus on Importinβ for simplicity and consider its surface footprint of roughly 20 nm$^2$ for the capsids with highest #NLS (1:1 stoichiometric complex of NLS and Importin), the overall surface shielding by Importins is roughly 100% for MS2$^{S37P}$, 80% for MS2 and 50% for I53-47. Thus, the substantial cargo decoration with Importins would result in a larger portion of the capsid surface being shielded.

## Discussion

Our approach based on modified capsids with tuneable surface properties and quantitative imaging in permeabilised cells enabled us to arrive at a substantially enhanced quantitative understanding of large cargo transport through the NPC. Assaying nuclear import kinetics in an unprecedented cargo size and #NLSs range, we have shown that the requirements for transport scale non-linearly with size and can be recapitulated by a two-parameter biophysical model that correlates the import flux to the energetic requirements for nuclear transport.

For small cargo transport, biochemical or physicochemical properties of the cargo surface have been shown to influence nuclear transport (*Frey et al., 2018*; *Naim et al., 2009*). While we do not claim that surface effects play no role in large cargo transport, based on the prediction from our experimental assay we would suggest that the binding of multiple Importin complexes seem to partially mask the cargo surface properties.

Our work significantly expands the range of sizes and #NLSs for which nuclear import has been characterised: *Tu et al., 2013* previously reported a single molecule study of a β-galactosidase cargo, which has four NLSs. This approximately cylindrical molecule is 18 nm at its longest axis, similar to MS2$^{S37P}$, and 9 nm along its shorter axis. If the cargo crosses the NPC in a favourable orientation (through its narrow end), this would result in a lower cost of insertion and explain well why for this substrate 4 NLSs are sufficient for import (*Tu et al., 2013*). By comparison, our smallest cargo, MS2$^{S37P}$, which is spherical with a 17 nm diameter, was not substantially imported below 10 NLSs within the timeframe of our assay. For the larger MS2, more than 30 NLSs were required to detect a clear signal. It is important to note that in addition to cargo shape (*Mohr et al., 2009*), its mechanical stability and rigidity are likely to play a role in nucleocytoplasmic transport: the import rate of proteins is inversely correlated with its mechanical stability (*Infante et al., 2019*), and flexibility is likely relevant for the transport of large deformable synthetic cargoes, such as polymer vesicles (*Zelmer et al., 2020*).

While our simple biophysical model can explain the experimental data very well with only two fitting parameters per capsid ($F(R)$ and $\epsilon$) it also raises several interesting questions. The model provides quantitative estimates of the free energy cost of capsid insertion into the FG Nup assembly, as well as the effective binding energy needed to compensate for the insertion cost. Notably, despite the fact that a single MS2 capsid already occupies $\approx 1/3$ of the estimated volume of the central NPC channel (*Isgro and Schulten, 2005*) (as illustrated in the cartoon in *Figure 5*) the free energy cost of insertion is relatively low (on the order of a few $k_BT$'s), and is similar for the capsids of different sizes. This might indicate that further mechanisms facilitate large cargo transport, such as NPC scaffold dilation, a hypothesis supported also by multiple evidences for tentative hinge elements in the NPC

scaffold structures (*Bui et al., 2013*; *Kelley et al., 2015*), or bistability in the FG density in the radial direction induced by such extremely large cargo (*Osmanović et al., 2013*).

In terms of the effective interaction energy $\epsilon$, the largest MS2 capsid required a fit with the lowest effective $\epsilon$. This finding is surprising at first glance, because one would expect that the main difference between the capsids would stem from their size difference, while the interaction energy of an NTR with an FG motif stays relatively constant. One can think of several potential origins for this effect, among those are the lack of independence in the NTR binding of the FG repeats in case of large surface coverage, or the loss of accessibility of the FG motifs due to the high compression of the FG assembly by the largest capsid, which will be explored in future work. Nevertheless, we found that all these features can be explained in a minimal model that incorporates the potential heterogeneity of the FG Nup distribution along the NPC axis, whereby there are at least two different regions of different FG Nup densities, as has been previously suggested in a 'two gate' or 'vestibule' pictures of the NPC (*Tu et al., 2013*; *Lowe et al., 2015*; *Yamada et al., 2010*). In *Figure 5—figure supplement 3*, we show that such a spatially heterogeneous model would be consistent with the data across all three capsid data sets, without invoking different effective interaction energies for the different capsids.

In our model, the energetic terms represent the binding between FG repeats and NTRs. The microscopic binding mechanism between NTRs and FG repeats during NPC passage is probably similar both for import and export, with a few exceptions - such as the binding of the export factor Crm1 to a specific stretch of Nup214, (which has likely a larger role in undocking than permeability barrier passage) (*Port et al., 2015*; *Tan et al., 2018*). We thus anticipate that basic principles of our work could also help in the future to better understand export of large cargoes, such as pre-ribosomal subunits and large RNA export complexes.

The theoretical model used in this paper implicitly assumes that the capsids do not interact with each other during transport through the pore. We cannot exclude multiple capsids colliding with each other in a single pore with absolute certainty - and this has indeed been observed in EM of HBV capsids injected in *Xenopus* oocytes (*Panté and Kann, 2002*). However, the hallmark of jamming resulting from multi-particle occupancy is the non-monotonic dependence of the flux on the interaction strength and thus on the #NLS on the capsid (*Zilman, 2009*; *Pagliara et al., 2013*) - a trend that is currently not apparent in our data within the experimental accuracy, at least in the initial rate, which is the focus of our analysis.

A more complete picture of nuclear transport and refined model building in the future would require taking into account additional features in more detail, such as docking and undocking from the barrier, more realistic modelling of the capsid cargo passage through the pore, and complex entropic effects of capsid-FG Nup interactions. Future studies using our cargo substrates and time resolved high-resolution measurements could provide further insights into the individual kinetic steps of NPC binding, barrier passage and undocking and how those link to FG Nup and potentially scaffold dynamics in the NPC.

## Materials and methods

**Key resources table**

| Reagent type (species) or resource | Designation | Source or reference | Identifiers | Additional information |
|---|---|---|---|---|
| Strain, strain background (*E. coli*) | BL21 | Invitrogen/Thermo Fisher Scientific | | AI strain |
| Cell Line (*Homo-sapiens*) | HeLa Kyoto | Gift from Martin Beck's Lab | RRID:CVCL_1922 | |
| Recombinant DNA reagent | pBAD_MS2_Coat_Protein –(1–393) (plasmid) | This study | | Protein expression plasmid for *E. coli* (MS2) |
| Recombinant DNA reagent | pET29b(+)_I53–47A.1–B.3_D43C (plasmid) | This study | | Protein expression plasmid for *E. coli* (I53-47) |

*Continued on next page*

*Continued*

| Reagent type (species) or resource | Designation | Source or reference | Identifiers | Additional information |
|---|---|---|---|---|
| Recombinant DNA reagent | pBAD_MS2_Coat_Protein–(1–393)_S37P (plasmid) | This study | | Protein expression plasmid for *E. coli* (MS2$^{S37P}$) |
| Recombinant DNA reagent | pET28a2-SCSG-GB1-coreN-GFP$\beta$1–10//NLS-GFP$\beta$11-coreC149H6 (plasmid) | This study | | Protein expression plasmid for *E. coli* (HBV SplitCore) |
| Recombinant DNA reagent | pBAD-MCS-CoreN-cys-loop-CoreC-TEV-12His (plasmid) | This study | | Protein expression plasmid for *E. coli* (HBV core with cysteine and NLS) |
| Recombinant DNA reagent | pET28a2-HBc14S H6_S81C (plasmid) | This study | | Protein expression plasmid for *E. coli* (HBV core with cysteine) |
| Recombinant DNA reagent | pTXB3-12His-Importin beta WT (plasmid) | This study | | Protein expression plasmid for *E. coli* (Impβ) |
| Recombinant DNA reagent | pBAD-Import$\alpha$1-FL-Intein CBD-12His (plasmid) | This study | | Protein expression plasmid for *E. coli* (Impα) |
| Recombinant DNA reagent | pTXB3-NTF2-intein -6His (plasmid) | This study | | Protein expression plasmid for *E. coli* (NTF2) |
| Recombinant DNA reagent | pTXB3-Ran Human FL-Intein-CBD-12His (plamid) | This study | | Protein expression plasmid for *E. coli* (Ran) |
| Peptide, recombinant protein | NLS peptide | PSL GmbH | | Mal-GGGGKTGRLESTP PKKKRKVEDSAS |
| Peptide, recombinant protein | NLS peptide with additional charges | PSL GmbH | | Mal-DEDED-GGGGKTGRLESTPP KKKRKVEDSAS |
| Chemical compound, drug | Hoechst | Sigma | B2261 | For nuclei labelling |
| Chemical compound, drug | Mitotracker green | Invitrogen | M7514 | For mitochondria labelling |
| Chemical compound, drug | FITC-Dextran 70 kDa | Sigma | 53471 | Used for checking nuclear envelope integrity |
| Chemical compound, drug | Alexa Fluor 647 maleimide | Invitrogen | A20347 | Dye for capsid labelling |
| Software, algorithm | UCSF Chimera | http://www.rbvi.ucsf.edu/chimera/ | RRID:SCR_004097 | |
| Software, algorithm | Fiji | https://fiji.sc/# | RRID:SCR_002285 | |
| Software, algorithm | SymphoTime | PicoQuant | RRID:SCR_016263 | |
| Software, algorithm | Igor Pro | Wavemetrics | RRID:SCR_000325 | |
| Software, algorithm | Adobe Illustrator CS6 | Adobe | RRID:SCR_010279 | |

## Large cargo expression and purification
### MS2 and MS2$^{S37P}$ capsids

A colony of *E. coli* BL21 AI cells containing the pBAD_MS2_Coat Protein-(1-393) or the pBAD_MS2_-Coat Protein-(1-393)_S37P plasmids was inoculated in LB medium containing 50 µg/mL ampicillin. The culture was grown overnight shaking at 37°C (180 rpm) and then used at a 1:100 dilution to inoculate an expression culture in LB medium. Protein expression was induced at OD$_{600}$ = 0.6–0.7 by

adding 0.02% arabinose and carried out at 37°C shaking (180 rpm), for 4 hr. Cells were harvested by centrifugation in a Beckmann centrifuge, rotor JLA 8.100 at 4500 rpm, for 20 min, at 4°C. For purification, pellets were resuspended in an equal volume of lysis buffer (10 mM Tris pH 7.5, 100 mM NaCl, 5 mM DTT, 1 mM $MgCl_2$, 1 mM PMSF) and lysed through 3–4 rounds in a microfluidizer, at 4°C. The lysate was incubated with 0.2% PEI (polyethylenimine) for 1 hr, on ice and then clarified by centrifugation in a Beckmann centrifuge, rotor JA 25.50 at 10,000 rpm, for 30 min. A saturated solution of $(NH_4)_2SO_4$ was added at 4°C drop-wise to the clear lysate under continuous mild stirring up to 25% of ammonium sulphate. After 1 hr, the lysate was spun down again at 10000 rpm, for 30 min. The supernatant was discarded, and the pellets were gently resuspended with 10–20 mL of lysis buffer on a rotator, at room temperature. The lysate was then centrifuged at 10,000 rpm, for 30 min and the clear supernatant was collected. The supernatant was cleared using the KrosFlo system (SpectrumLabs) with a 0.2 µm cut-off membrane to remove large impurities. The membrane permeate containing the cleared sample was collected on ice. In order to maximise protein recovery, the remaining supernatant was washed with 50 mL of lysis buffer and the permeate was pooled with the previously collected one. The sample was then concentrated using the KrosFlo with a 500 kDa cut-off membrane (for the smaller MS2$^{S37P}$ capsid, a 30 kDa cutoff was used).

## I53-47 capsids

A colony of *E. coli* BL21 AI cells containing the pET29b(+)_I53-47A.1-B.3_D43C plasmid was inoculated in LB medium containing 50 µg/mL kanamycin. The culture was grown overnight shaking at 37°C (180 rpm) and then used at a 1:100 dilution to inoculate an expression culture in LB medium. Protein expression was induced at $OD_{600}$ = 0.8 by adding 1 mM IPTG and carried out at 37°C shaking (180 rpm), for 3 hr. Cells were harvested by centrifugation in a Beckmann centrifuge, rotor JLA 8.100 at 4500 rpm, for 20 min, at 4°C. The purification procedure was adapted from *Bale et al., 2016*. Pellets were resuspended in two pellet volumes of lysis buffer (25 mM Tris pH 8.0, 250 mM NaCl, 20 mM imidazole, 1 mM PMSF, 0.2 mM TCEP), sieved to remove clumps and supplemented with 1 mg/mL lysozyme and DNase. Cells were lysed by sonication on ice, and the lysate was clarified by centrifugation at 24,000 g, for 35 min, at 4°C. The clear lysate was incubated with Ni-beads (1 mL/L expression) for 1–2 hr, at 4°C under gentle rotation. Ni-beads with lysate were poured in a polypropylene (PP) column and the flow through (FT) was collected. Ni-beads were washed three times with 20 mL of lysis buffer followed by elution with 5 mL of elution buffer, containing 500 mM imidazole. The elution was immediately supplemented with 5 mM EDTA to prevent Ni-mediated aggregation of the sample. The buffer of the protein was then exchanged to dialysis buffer (25 mM Tris pH 8.0, 150 mM NaCl, 0.2 mM TCEP), at 4°C. After dialysis, the protein was transferred to a new tube and spun down for 10 min, at 5000 rpm, at 4°C, in order to remove any precipitation. The protein was concentrated using the KrosFlo with a 100 kDa cutoff membrane, which also helps removing any remaining unassembled capsid proteins. After concentrating down to 3–4 mL of volume, the sample was washed with 50 mL of fresh dialysis buffer using the continuous buffer exchange mode of the KrosFlo.

## HBV capsids

A colony of *E. coli* AI cells containing the desired HBV plasmid was inoculated in TB medium containing 50 µg/mL ampicillin. The culture was grown overnight shaking at 37°C (180 rpm) and then used at a 1:100 dilution to inoculate an expression culture in LB medium. Protein expression was induced at $OD_{600}$ = 0.8–1 by adding 0.02% arabinose and carried out at 20°C shaking (180 rpm) overnight. Cells were harvested by centrifugation in a Beckmann centrifuge, rotor JLA 8.100 at 4500 rpm, for 20 min, at 4°C. The purification procedure was adapted from *Walker et al., 2011*. Pellets were resuspended in one pellet volume of lysis buffer (25 mM Tris pH 7.5, 500 mM NaCl, 0.2 mM TCEP, 10 mM CHAPS) and lysed by sonication 3 × 30 s, on ice. The lysate was spun down in a Beckmann centrifuge rotor JA 25.50 at 10,000 rpm, for 10 min. The cleared supernatant was then loaded on a step gradient 10–60% sucrose obtained by mixing lysis and sucrose buffers (25 mM Tris pH 7.5, 500 mM NaCl, 0.2 mM TCEP, 10 mM CHAPS, 60% sucrose) in appropriate ratios and by carefully layering the different percentage buffers into ultracentrifugation tubes. The lysate was then subjected to ultracentrifugation at 28,000 rpm, for 3.5 hr at 4°C. Fractions of 2 mL were collected by gravity, by

puncturing the ultracentrifugation tube from the bottom. Fractions containing the capsids were pooled and concentrated using the KrosFlo with a 500 kDa cutoff membrane.

## Large cargo maleimide labelling and characterisation

Purified capsids were labelled via maleimide chemistry to couple a fluorescent dye and NLS peptide to the exposed cysteines. The dye (AlexaFluor647 maleimide, Invitrogen) and NLS peptide (Maleimide-GGGGKTGRLESTPPKKKRKVEDSA, PSL Peptide Specialty Laboratories) were stored at −80°C and freshly resuspended in anhydrous DMSO. The capsids were incubated with different molar excesses of dye and NLS peptide according to the desired degree of labelling for 1–2 hr, at room temperature. A typical reaction was: 30–50 nmol of protein, 50 nmol of dye and 100–250 nmol of NLS peptide depending on the target #NLSs. The reaction was then quenched by adding 10 mM DTT and the protein was spun down at 10,000 rpm, for 10 min, at 4°C to remove any precipitation. The excess dye was removed by loading the capsid sample on a HiPrep Sephacryl 16/60 size exclusion column (GE Healthcare), using the appropriate buffer (for MS2 and MS2$^{S37P}$: 10 mM Tris pH 7.5, 100 mM NaCl, 5 mM DTT; for I53-47: 25 mM Tris pH 8.0, 150 mM NaCl, 1 mM DTT and for HBV: 25 mM Tris pH 7.5, 500 mM NaCl, 0.2 mM TCEP, 10 mM CHAPS, 10% sucrose). Relevant fractions containing the labelled capsids were then pooled and concentrated using the KrosFlo. For long-term storage at −80°C, the sample was supplemented with 25% glycerol (30% sucrose for HBV) and either flash-frozen with liquid nitrogen or directly transferred to the −80°C freezer (for I53-47 capsids). The ratio of capsid monomers tagged with NLS peptide was quantified by the gel band ratio on a SDS PAGE gel with Coomassie staining, as the labelled monomers migrate differently due to their increased molecular weight. We note that the quantified #NLSs represents an average of the labelling reaction.

## Electron microscopy

Capsid integrity was confirmed by imaging the samples with an electron microscope using negative staining. Carbon-coated 300 meshes Quantifoil Cu grids were glow-discharged for 10 s in a vacuum chamber. Then, a 3 μL drop of sample was adsorbed on a grid for 2 min, blotted with Whatman's filter paper and washed three times with sample buffer, then three times with a solution of 2% uranyl acetate. Once the grids were dry, the sample was imaged using a Morgagni 268 microscope (FEI).

## Dynamic light scattering

Dynamic light scattering (DLS) measurements to quantify the hydrodynamic radius of capsids and test for sample aggregation or disassembly were performed on a Zetasizer Nano (Malvern). Samples were diluted to a final concentration of 0.5 μM in filtered TB and spun down for 10 min at 10,000 g prior to each measurement. For each sample, at least 10 measurements were acquired, using a 12 μL quartz cuvette. Count rates per second were typically higher than 200 kcps, and the polydispersity index was below 0.2, indicating a monodisperse solution. Data were analysed using the Malvern software, using the Multiple Narrow Bands fitting algorithm and Refractive Index and Absorption settings for proteins.

## Fluorescence correlation spectroscopy

Fluorescence correlation spectroscopy (FCS) was used to characterise the large cargoes and quantify their concentration and brightness (#dyes/capsid). FCS experiments were carried out on a custom-built multiparameter spectrometer confocal setup, equipped with a 60x water objective (NA = 1.27). The capsid samples were diluted in freshly filtered 1XTB and spun down for 10 min at 10.000g at 4°C prior to the start of the experiment. FCS measurements were carried out in 8-well Lab-Tek, which had been pre-incubated for 30 min with a solution of 1 mg/ml BSA to prevent sample sticking. For each sample, at least 10 FCS curves of 30 s each were acquired. Low power (1–5 μW) was used to avoid bleaching of the samples during their diffusion through the confocal volume. A calibration FCS measurement with a free dye solution was carried out every 2–3 samples to measure the structural parameter and confirm the stability of the setup. Data analysis was performed with SymphoTime software. Autocorrelation curves were computed for lag times between 0.0001 and 1000 ms and fitted with a diffusion model. Capsid brightness was calculated by dividing the measured particle

brightness by the measured brightness of a calibration dye solution at the same laser power settings. Due to large aggregates in the absence of Importins, HBV was not probed by FCS.

## Nuclear import assays

HeLa Kyoto cells were cultured at 37°C, 5% $CO_2$ atmosphere in Dulbecco's modified Eagle's medium with 1 g/mL glucose (Gibco 31885023) supplemented with 1% penicillin-streptomycin (Sigma P0781), 1% L-Glutamine (Sigma G7513) and 9% FBS (Sigma F7524). The cells were regularly tested for mycoplasma contamination and found to be mycoplasma-negative. The cells were passaged every 2–3 days up to maximum of 15–17 passages. Cells were seeded 1 or 2 days prior the experiment at low density (10,000–12,000 cells per well) in a glass-bottom eight-well Lab-Tek II chambered coverglass (Thermo Scientific Nunc, 155383).

Cells for transport assays were stained with 100 nM MitoTracker green (Invitrogen, M7514) in growth medium for 30 min, at 37°C, 5% $CO_2$. For nuclear staining, cells were rinsed once with PBS and incubated for 10 min, at room temperature with 20 nM Hoechst 33342 (Sigma, B2261).

Cells were then washed once with transport buffer (1XTB: 20 mM Hepes, 110 mM KOAc, 5 mM NaOAc, 2 mM MgOAc, 1 mM EGTA, pH 7.3 adjusted with KOH) and permeabilised by incubation for 10 min, at room temperature with digitonin (40 µg/mL). Cells were then washed 3 times with 1XTB supplied with 5 mg/mL PEG 6000 to avoid osmotic shock. After the final wash, excess buffer was removed and the transport mix was quickly added to the cells to start the experiment. The final transport mix was composed of 1 µM Importinα, 1 µM Importinß, 4 µM RanGDP, 2 µM NTF2, 2 mM GTP and 8 nM capsid cargo. In order to allow the import complex to form, the cargo was first pre-incubated with Importinß and Importinα on ice for at least 10 min, then the rest of the transport mix was added and the solution was spun down for 10 min at 10000 g to remove any aggregates. Each experiment was performed side-by-side with control cells incubated with fluorescently labelled 70 kDa Dextran (Sigma 53471) to confirm nuclear envelope intactness throughout the whole experiment. We note that, in order to exclude possible contaminations of free capsid monomers and/or fragments in our experiments, we applied stringent quality checks to each capsid prep and only used samples that had all of the following: uniform EM images, good quality monodisperse DLS, FCS parameters in line with monodisperse cargo of the right size.

## Microinjection in starfish oocytes

Starfish (*Patiria pectinifera*) were kindly provided by Kazoyushi Chiba (Ochanomizu University, Tokyo, Japan) and kept at 16°C in seawater aquariums at MPI-BPC's marine facilities. Oocytes were extracted from the animals fresh for each experiment as described earlier (*Lénárt et al., 2003*). Fluorescent proteins were injected using microneedles, as described previously (*Borrego-Pinto et al., 2016*; *Jaffe and Terasaki, 2004*).

## Confocal fluorescence microscopy

Time-lapse confocal imaging of nuclear import was performed on an Olympus FLUOVIEW FV3000 scanning confocal microscope, using a 40x air objective (NA = 0.95). An automated multi-position acquisition was carried out, where 12 different regions (typically containing 10 cells each) were imaged in two different wells. Three channels were recorded at each time step, using the 405 nm (Hoechst), 488 nm (Mitotracker) and 640 nm (cargo) laser lines for excitation. Images were acquired every 2 min for 80–90 min, using continuous autofocusing with Z-drift compensation to ensure imaging stability.

## Image and data analysis

Results of the time-lapse import experiments were analysed with a custom-written Fiji (*Schindelin et al., 2012*) script (*Source code 1*). The Hoechst and Mitotracker channels were used to generate reference masks for the nucleus, nuclear envelope and cytoplasm at each time point. Briefly, the two images were pre-processed with Gaussian blur to aid in area segmentation, and then thresholded. The nuclear mask was eroded three times to remove contributions coming from the nuclear envelope, and the envelope mask was generated by subtracting the eroded mask from the non-eroded one. The final masks were then used to extract the average intensity of cargo signal in the different areas of interest. Final data analysis and plotting was performed in IgorPro

(Wavemetrics). Fluorescence intensities were background-corrected, rescaled according to the capsid brightness and fitted to an inverse exponential function $I(t) = A + I_{MAX}(1 - e^{-\tau * t})$, with $I_{MAX}$ being the plateau value reached at infinity, $\tau$ the reaction constant with units 1/s and $A$ an offset parameter.

### Mathematical analysis of the data

The initial flux was estimated from the mono-exponential fit as $J = \tau \cdot I_{MAX}$. Error bars in the initial flux show sample standard deviations across the 12 imaged regions. For comparison with the theoretical predictions, where the flux saturates to a maximal value $J_{max}/(k_{ON}c) = \frac{1}{a_{Ran}}$ reached as $N \to \infty$, we normalised all the flux measurements by the maximal observed value of the flux across all the technical replicate (that was still within 95% confidence interval of the mean value). Changing the normalisation value of the flux does not qualitatively change the conclusions of the model; however, it may cause a slight increase to our $F(R)$ values and a slight decrease to our $\epsilon$ values. $F(R)$ and $\epsilon$ values were obtained using a least-squares fit implemented in Python. Plots of $\Delta G$ and overlays of our fits onto the initial flux were also performed using Python.

## Acknowledgements

We are particularly grateful to Niccolò Banterle for help with initial HBV experiments. We are very grateful to Peter Lenart and the light microscopy facility at the MPI for Biophysical Chemistry for their significant help with the oocyte work. We also thank Gemma Estrada Girona, Miao Yu and Sofya Mikhaleva for their contributions to this work. We thank SFB 1129 (Project number 240245660 funded by DFG, German Research Foundation) and the National Science and Engineering Research Council of Canada (NSERC) for generous funding. We thank the Baker lab for providing plasmids for their artificial capsid structure and the Nassal lab for HBV constructs. We thank Ulrich Schwarz for insightful discussion. We also thank the ALMF and EM facilities as well as the mechanical workshop at the EMBL.

## Additional information

### Funding

| Funder | Grant reference number | Author |
|---|---|---|
| Deutsche Forschungsgemeinschaft | 240245660 | Giulia Paci<br>Joana Caria<br>Edward A Lemke |
| Natural Sciences and Engineering Research Council of Canada | | Tiantian Zheng<br>Anton Zilman |

The funders had no role in study design, data collection and interpretation, or the decision to submit the work for publication.

### Author contributions

Giulia Paci, Software, Formal analysis, Validation, Investigation, Visualization, Methodology, Writing - original draft, Writing - review and editing; Tiantian Zheng, Software, Formal analysis, Validation, Investigation, Writing - review and editing; Joana Caria, Validation, Investigation, Methodology, Writing - review and editing; Anton Zilman, Resources, Supervision, Funding acquisition, Investigation, Methodology, Writing - review and editing; Edward A Lemke, Conceptualization, Resources, Supervision, Funding acquisition, Investigation, Methodology, Writing - original draft, Project administration, Writing - review and editing

### Author ORCIDs

Giulia Paci ![ORCID] https://orcid.org/0000-0003-0565-4356
Anton Zilman ![ORCID] https://orcid.org/0000-0002-8523-6703
Edward A Lemke ![ORCID] https://orcid.org/0000-0002-0634-0503

Decision letter and Author response
Decision letter https://doi.org/10.7554/eLife.55963.sa1
Author response https://doi.org/10.7554/eLife.55963.sa2

# Additional files

## Supplementary files

• Source code 1. Source ImageJ/Fiji macro to measure nuclear intensities. The code can be executed in either Fiji or ImageJ. It uses the two reference channels (nuclear and mitochondrial staining) to segment the nucleus, nuclear envelope and cytoplasm and measure the cargo fluorescence intensity in these regions for each frame.

• Supplementary file 1. Comparison with bi-exponential fit. We evaluated the appropriateness of a single- vs double-exponential fit to our kinetic data. In this file we report the fit parameters for the double exponential fit with their very high uncertainties and show that their combinations is tightly constrained to values of the mono-exponential fit parameters.

• Transparent reporting form

## Data availability

All data generated or analysed during this study are included in the manuscript and supporting files. Raw image data is available via IDR (https://idr.openmicroscopy.org/search/?query=Name:87).

The following dataset was generated:

| Author(s) | Year | Dataset title | Dataset URL | Database and Identifier |
|---|---|---|---|---|
| Paci G, Zheng T, Caria J, Zilman A, Lemke EA | 2020 | Molecular determinants of large cargo transport into the nucleus | https://idr.openmicroscopy.org/search/?query=Name:87 | Image Data Resource, idr0087 |

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
