## [Decision Letter]

**Acceptance summary:**

The requirements for nuclear transport can be recapitulated by a simple two-parameter biophysical model that correlates the import flux with the energetics of cargo transport through the nuclear pore complex. Together, the results reveal key molecular determinants of large cargo nuclear import in cells.

**Decision letter after peer review:**

[Editors’ note: the authors submitted for reconsideration following the decision after peer review. What follows is the decision letter after the first round of review.]

Thank you for submitting your work entitled "Molecular determinants of large cargo transport into the nucleus" for consideration by *eLife*. Your article has been reviewed by a Reviewing Editor and a Senior Editor, a Reviewing Editor, and two reviewers. The reviewers have opted to remain anonymous.

Our decision has been reached after consultation between the reviewers. Based on these discussions and the individual reviews below, we regret to inform you that your work will not be considered further for publication in *eLife*.

While the reviewers felt the work had merit, they had several concerns about the interpretation of the data, the limitations of the experiments and the relevance to living cells. Their concerns might be addressed by further experiments but in its present state we feel the manuscript has too many issues to address within a few months. Hopefully their comments will be helpful in improving your manuscript. When you have addressed the concerns, we could consider it for *eLife*, as a new manuscript.

Reviewer #1:

Compared with typical/average-sized nuclear transport cargos, there are little data on the transport rates of large cargos by nuclear pore complexes (NPCs). The authors examined the suitability and transport rates of a series of 5 viral particle capsids of different sizes and with different numbers of NLSs. They conclude that 10 or more NLSs are required for import for the cargos examined, and that the number of NLSs per cargo volume is a better variable than NLSs per cargo surface area for predicting the amount of nuclear uptake. While these are interesting and potentially useful quantitative results, there are significant issues with the results, interpretation and details provided, which tempers my enthusiasm.

Of the 5 viral particle capsids described, the authors had technical difficulties with the two largest, and therefore these were not included in the analysis. Thus, their abstract is misleading as they do not report kinetics for the size range of 17-36 nm, but rather 17-27 nm. The problems with the two largest capsids should be moved to a supplementary section so as not to distract from the main work. In addition, the MS2 cargo is so poorly imported that it does not make sense to use it to draw major conclusions. For example, the slope for MS2 in Figure 5A is so flat that it is impossible to reliably conclude a minimum number of NLSs. The remaining cargos are not well behaved in terms of transport kinetics, as described in more detail below.

The NLSs are randomly attached to the capsid surfaces, making the resultant populations heterogeneous. The quantified number of NLSs is an average, and gel analysis is semi-quantitative. Some discussion of their expected errors is warranted.

The only biologically relevant capsid (HBV) – i.e., one that is imported into the nucleus – is not included in their analysis. Physiologically, this capsid disassembles in the nuclear basket. They have deleted the authentic NLS for their experiments. Thus, the biological implications are limited.

In Figure 1D, are the curves actual data or fits? FCS and DLS signals will be dominated by the large particles, yet free dye and/or labeled capsid monomers can significantly influence the import curves – are these responsible for the non-zero ordinate intercepts in Figure 3—figure supplement 1? Can the labeled capsids be separated from monomers and free dye by size exclusion chromatography?

The authors discount the importance of surface properties at numerous locations throughout the text. But they have not actually tested this, and surface properties are in fact surprisingly important, as multiple studies have shown – changing a few residues or adding fluorescent dyes can dramatically change the import properties of cargos. In fact, I would not be surprised if varying the number of dyes on their cargos would alter the slopes of the plots in Figure 5, or some of the scatter in these plots arises from the dye:cargo:NLS ratio. Minimally, they should tone down their discussion arguing against a minimal influence of surface properties.

While the authors limit the fitting "to the first 40 minutes to extract more accurate kinetics", the opposite is in fact true. Accurate fitting of exponential kinetics requires knowing the asymptotic limit, which is not the case for numerous curves in Figure 3 -figure supplement 1. Also, initial time points in these curves vary widely – this is not expected or discussed.

For 80 min time points, the authors should really consider including CAS, RanGAP and RanBP1 to maintain complete recycling of transport factors.

"Normalized nuclear intensity" needs some explanation. Relative to what? Do these correspond to the same scale for different plots. What does an intensity of 1 signify? How does this relate to the intensity in Figure 1D? The efficiency of nuclear uptake of the different cargos varies widely, but this is not discussed.

The energetic discussion in the last paragraph has little meaning without an estimate of the entropic cost of displacing the permeability barrier.

Reviewer #2:

The manuscript by Paci and Lemke describes experiments addressing nuclear accumulation of large NLS-labeled cargoes. The effort is commendable and the use of modified viral capsids is admirably clever. However, I have some serious problems with the interpretation.

The experiments are based on permeabilized cell assays. These are standard in the field, for better or worse, but they suffer a generic problem in that the rest of the cell is washed away. In a live cell, the transport substrate of interest has to compete with the rest of the proteome for attentions of the transport receptors. This can have a dramatic effect on the transport kinetics.

Like most studies of nuclear accumulation, the analysis does not distinguish properly between permeability of the nuclear envelope and the saturating level of nuclear concentration. The latter is recognized as "robust nuclear import" but depends, quite obviously, on the RanGTP system. The assumption that monoexponential (first-order) kinetics measure permeability through the nuclear pores is simply not justified. The observed kinetics reflect the rate-limiting step, which may be Ran recharging with GTP or recycling to the cytoplasm. See Kim and Elbaum, 2013, and much earlier Smith et al., 2002.

Quantitative measurements of nuclear accumulation can be affected in addition by binding to structures within the nucleus, as suggested by the images in Figure 3 for MS2 with high NLS count. Each NLS adds a considerable amount of positive charge. This may well affect binding to nucleic acids when present in such high local concentration on the viral capsid, especially if DNA/RNA binding proteins are lost in the permeabilization.

The text deals with the level of nuclear accumulation ("endpoint" in Figure 5), but the graphs presented show the accumulation kinetics rather than the saturation as a function of #NLS. The time for half-saturation, (I(t) – A)/Imax = 1/2, is actually ln2/k, not ln2/Imax as written in the text (subsection “Image and data analysis”). Looking at the table in Supplementary file 1, the values for T_1/2 are listed equal to 1/2 * ln2/k. This has the correct units but I don't understand the factor of 1/2.

If the aim of the exercise is to study the degree of accumulation, i.e., Imax, then the proper parameter to measure is the saturating nuclear to cytoplasmic ratio N:C. The logarithm of this ratio is the chemical potential difference, which is the essential thermodynamic quantity. As presented, the data do not show the cytoplasmic intensity and the background correction that was applied is not described. Figure 2C shows a single example of the cytoplasmic intensity where the nuclear to cytoplasmic ratio saturates at about 10 (700 / 70 units on the graph).

Since the fluorescence external to the cells coming from titrated cargo substrates should equilibrate with the fluorescence in the cytoplasm, I looked to see if this might be included in the fitting parameter A. It was not clear whether A is the background correction itself or a fit after the correction is applied. In any case A cannot represent the fluorescence from free cargo. According to the text these are introduced at a constant 8 nM concentration, but the values listed in the supplementary file vary widely, even for a given class of cargo. Why should they vary so widely? Presumably these values are corrected by the same factor as Imax for the substrate brightness. If they are not corrected, shouldn't the capsids with fewer NLS appear brighter, so with larger A? In some cases A is a very large fraction of Imax, leaving little dynamic range for the measurement itself. (Compare I53-47 with 15, 18, and 22 NLS.) In principle the black level to subtract is that of the confocal microscope with the laser blocked, and the fluorescence in the surrounding medium should match that measured in the permeabilized cytoplasm. If the cells are auto-fluorescent in the measurement channel then some additional correction will be required, but it should be specified clearly.

A few relatively technical points:

Why was the labeling with fluorescent dye and NLS done both on cysteine? The proteins could have been labeled first on lysine and then with NLS on the cysteine. The problem is that the molecular weight of the dye is almost half that of the peptide. Is a control available to show that the dye labeling really has no effect on the gel mobility? Figure 1—figure supplement 1 shows both Coomassie and fluorescence in the "unsuccessful" labeling of I53-50. For clarity, the main figure should also show the fluorescence in the successful case.

I did not understand the toy model in subsection “Global quantitative analysis of nuclear import in relation to cargo size and #NLSs”. The binding energy of NTRs to the cargo does not assist in directional translocation, nor is it transferred to displacing the FG repeats. That depends on interactions of NTRs with FG motifs. Crowding in the nuclear pore as shown in Figure 5 is interesting and might relate to kinetics, but not to the saturating concentration ("endpoint").

Nuclear export is not just the inverse of import. See Kim and Elbaum, 2013. There is a fundamental difference between exchange of RanGTP, a reversible reaction in "import", and physiologically irreversible GTP hydrolysis, which is coupled to translocation in "export".

The manuscript is long for a short report, about 3500 words in the main text alone.

Hoping to end on a constructive note, I have to apologize for being such an ornery reviewer here. I do quite like the experiment and I believe the data hold some new truths to be discovered. Wherever the work is ultimately published, I would like very much to see the nuclear accumulation presented as the nuclear to cytoplasmic ratio. This will normalize inherently for substrate brightness and avoid potential inconsistencies carried in by numbers from other measurements, imprecise dilutions, protein losses in aggregation, etc. Surely the data are available without requiring any further experiments. I am sure they could be reanalysed easily, avoiding confusion between kinetics and saturation. Plotting the ratio will clarify whether the additional number of NLS indeed influence the kinetics and saturation as suggested. There might be surprises in store.

[Editors’ note: further revisions were suggested prior to acceptance, as described below.]

Thank you for submitting your article "Molecular determinants of large cargo transport into the nucleus" for consideration by *eLife*. Your article has been reviewed by the original reviewer, and the evaluation has been overseen by a Reviewing Editor and Suzanne Pfeffer as the Senior Editor. The reviewer has opted to remain anonymous.

The Editors have discussed the review with one another and the Reviewing Editor has drafted this decision to help you prepare a revised submission.

One of the original reviewers feels the manuscript has been improved but has some issues with the interpretation of the data, and the model. Specifically, the reviewer states "Particular attention must be made to predictions of the model, and interpretations in the context of this model." This reviewer has been thorough in the evaluation, so we feel the comments may likely be helpful in improving the manuscript further.

Because the concerns can be answered without additional data, but only require revisions to the manuscript, or explanations for the reviewer, we opt to send it back to you to address these comments.

Reviewer 1:

This revised manuscript has been substantially improved by tightening up the discussion and presentation to focus on the main story, and with the addition of a mathematical model. However, I do have some concerns about the revised manuscript, listed below in order of importance. While some of these points address accuracy and a logical consistency, other portions are intended to promote a more nuanced and informative picture. Particular attention must be made to predictions of the model, and interpretations in the context of this model.

1) Figure 5B – The model impressively explains the values in the graph. However, all of the ∆G values are positive, suggesting that binding to the permeability barrier is unfavorable. Nonetheless, nuclear rimming is clearly seen during the import experiments, indicating that interaction with the pore is favorable – more favorable than being in the cytoplasmic compartment. This indicates that the NPC is a thermodynamic sink. The data thus seem incongruent with the model, which only postulates an energy barrier. The model in Figure 5—figure supplement 4 is reminiscent of the vestibule model of Tu et al., 2013), yet here too, none of the ∆G values are negative (which was the case in Tu et al.,). Please discuss.

2) They cite four references for the initial flux equation (2, 33-35). I cannot find the equation they use in these references. In fact, two of them describe flux in terms of a constant multiplied by a concentration difference, which seems inconsistent with their equation. More discussion is necessary to elucidate where the model comes from.

3) If I understand the methods correctly, the NLSs and dyes were simultaneously mixed with the capsids. They discuss tuning the NLS/capsid ratio, and this is ultimately determined via a gel shift assay. But what about the number of dyes per capsid? It seems like they have brightness data from FCS experiments, and this should be reported. Do the number of dyes vary inversely with the number of NLSs? They continue to minimize the role of surface properties, yet a few extra dye molecules were shown by Tu et al., to dramatically affect the permeability properties of the cargo. I do not consider it safe to assume that the number of dye molecules does not influence the particle's interaction strength with the NPC. Moreover, they state that F(R) scales with the radius, yet the values for F(R) that they obtain are all essentially the same, which would be consistent with different surface properties of the different diameter capsids. Stating this does not diminish their results.

4) The epsilon values are surprisingly small. For the cargo of Tu et al., this would predict a very small interaction strength of the fully decorated cargo, and even smaller for a single NTR-bound cargo, which nonetheless still clearly binds to the pore. Note that the size (volume occupied) of β-galactosidase is less than MS2(S37P) by a similar ratio that the MS2(S37P) size is less than I53-47. It would be quite surprising indeed if the substantial behavioral differences of the β-galactosidase and MS2(S37P) cargos can be ascribed to the size and shape differences alone. It seems that surface properties must at least somewhat contribute to the observed differences.

5) Discussion section – I do not understand these surface coverage calculations. For maximum NLSs of 38, 35, and 98 for MS2(S37P), I53-47, and MS2, I get 84%, 42%, and 85% surface coverage assuming 20 nm^2^/β. This does not include Importin α. How much do the diameters increase assuming a full coat of Importins α and β? This is expected to be significant. How does this increased diameter compare with the size of the channel? Is there any experimental evidence that all NLSs on the capsids are bound to NTRs? Taking into account that concentrations and the Kd (~40 nM, α for NLS) are similar, the NLSs on the MS2 capsid are only about 90% occupied, implying 77% surface coverage. While these changes may not materially change their interpretation, a more detailed discussion is necessary to build an accurate picture and to build confidence in the conclusions. Other potential complications: (1) is it possible geometrically for all NTRs on a capsid to be bound to FG repeats? Figure 5A suggests that this may not be possible; and (2) can multiple capsids simultaneously bind to a single pore? Excess cargo, slow import and nuclear rimming suggest this possibility. Would this affect interpretation?

6) It is unclear whether there is any meaning behind the A values. These are highly variable, and I don't know what to make of them. In principle, A could reflect the accumulation of the cargos on the nuclear envelope, but as this arises from an extrapolation to zero time, it seems like this should in fact be zero, or at least some reasonably explained value. One possibility is that import rate could be dependent on the amount of accumulated cargo at the pores, i.e., a release rate, as entrance into the NPCs appears really fast.

7) The data on negatively charged linkers is inconclusive at best, as they are highly scattered. Their conclusions should be toned down.

[Editors’ note: further revisions were suggested prior to acceptance, as described below.]

Thank you for resubmitting your work entitled "Molecular determinants of large cargo transport into the nucleus" for further consideration by *eLife*. Your revised article has been evaluated by Suzanne Pfeffer (Senior Editor) and a Reviewing Editor.

The manuscript has been improved but there are some remaining issues that need to be addressed before acceptance, as outlined below:

One reviewer feels the manuscript is substantially improved but there remains an outstanding issue that has not been corrected in the revision. The reviewer feels that Figure 5—figure supplement 4 needs to be clarified as described below. Additional minor comments directed at improving the manuscript are included as well. Please send a revised manuscript that addresses these comments sufficiently that it may not need to go back to this reviewer.

This revised manuscript has been substantially improved, with a much more balanced and informed discussion. All of my major concerns have been adequately addressed, with the exception of one item, the model in Figure 5—figure supplement 4. The figure itself is confusing/unclear, and I do not understand the basis behind building the model the way they did. Specific concerns for this figure are as follows:

1) What is the y-axis in the top panel of 'A'? This should be marked. My guess is that this is some measure of 'FG-Nup density' – are there any relevant units?

2) The dimensions of L1 and L0 do not reflect the values in the caption. Consequently, the diagram is misleading. The Greek letter is inconsistent with the caption. The vestibule region is not marked.

3) It is unclear why a transition region (L1) is included between the vestibule and L0. Comparing the top and bottom panels in A, it appears that the vestibule is equivalent to the cytoplasm. This does not make sense.

4) For L1 = 30 nm and L0 = 5 nm, the first impression is that the barrier gate is biased toward the nucleoplasmic side. Is this the intention? Such a model would be consistent with the nucleoplasmic gate hypothesized by the Weis group, and, if so, should be mentioned. Alternatively, are both the cytoplasmic and nucleoplasmic L1 regions both 30 nm? This would place the barrier in the center, but very narrow. It doesn't make much sense for a 'transition region' to be 6 times the width of the main barrier, so some discussion is needed here.

5) It is unclear why the ΔG for the L1 region changes substantially for the different viral particles, yet the ΔG for the L0 region changes minimally. It seems that the ΔG for the more dense FG nup environment would be more sensitive to particle size. An older hypothesis suggested dense clouds on the nucleoplasmic and cytoplasmic sides, but significantly lower density within the center. Is this being considered here?

6) The authors are correct in their rebuttal that only a portion of the NPC needs to contain a region where the interaction free energy is negative, in order to be consistent with the experimental observation of rimming. However, none of the regions illustrated in Figure 5—figure supplement 4 have negative ΔG. There is a dashed region that is apparently of negative free energy, but what this is remains unclear (point 3), and it is not clear if this energy is included in any way in their fit to the data.

7) In the lower panel of B, the green curve fit approximates the data very poorly, but does much better in the upper panel. Something seems amiss here.

---

## [Author Response]

[Editors’ note: the authors resubmitted a revised version of the paper for consideration. What follows is the authors’ response to the first round of review.]

Major revisions summary:

We thank both reviewers for their feedback on our work. Following up on all reviewers’ suggestions, we have substantially reworked the manuscript and expanded several points with new supporting data (both experimental and theoretical). We summarize below the main aspects before delving into a more specific point-by-point discussion of the individual reviewer comments:

1) We have completely changed the way we analyze the data, focusing on a more robust analysis of the measured kinetic parameters (completely revised Figure 5, and Supplementary file 2 and Supplementary file 3). This analysis has been performed in a new collaboration with the theoretical group of Dr. Anton Zilman at the University of Toronto. We now present a minimal biophysical model of large cargo transport through the NPC that only incorporates the most salient features of the energetics and the kinetics of transport. The model describes our experimental data very well and enables us to make inference regarding the major determinants of capsid transport in terms of their size and NLS number. We are indebted to the reviewers for encouraging us to look for a deeper interpretation of our data, resulting in a much improved paper that is now a well rounded blend of experiment and theory.

2) We have collaborated with the group of Dr. Peter Lenart at the MPI for Biophysical Chemistry Goettingen to perform injection experiments into live starfish oocytes (new Figure 2—figure supplement 2). All results agreed qualitatively with our permeabilised cell data. In addition, we want to note the merits of permeabilised cell work, which become evident when considering both experiments together: it enables us to put our results into a much larger context of available literature, as this is still the most frequent reported transport assay. Notably, the permeabilized cell assays also provided better access to the quantification of the biophysical parameters.

3) In addition, we reinforce our previous discussion of the role of the surface charges of the capsids with new, direct experimental data using capsids with modified surface charges (revised Figure 5C).

Reviewer #1:Compared with typical/average-sized nuclear transport cargos, there are little data on the transport rates of large cargos by nuclear pore complexes (NPCs). The authors examined the suitability and transport rates of a series of 5 viral particle capsids of different sizes and with different numbers of NLSs. They conclude that 10 or more NLSs are required for import for the cargos examined, and that the number of NLSs per cargo volume is a better variable than NLSs per cargo surface area for predicting the amount of nuclear uptake. While these are interesting and potentially useful quantitative results, there are significant issues with the results, interpretation and details provided, which tempers my enthusiasm.Of the 5 viral particle capsids described, the authors had technical difficulties with the two largest, and therefore these were not included in the analysis. Thus, their abstract is misleading as they do not report kinetics for the size range of 17-36 nm, but rather 17-27 nm. The problems with the two largest capsids should be moved to a supplementary section so as not to distract from the main work. In addition, the MS2 cargo is so poorly imported that it does not make sense to use it to draw major conclusions. For example, the slope for MS2 in Figure 5A is so flat that it is impossible to reliably conclude a minimum number of NLSs. The remaining cargos are not well behaved in terms of transport kinetics, as described in more detail below.

We thank the reviewer for the feedback but respectfully disagree with a few points: While the MS2 capsid shows much lower nuclear import total intensity when compared to the smaller MS2_S37P and I53-47 capsids, the signal we measure is still very well quantifiable, significant and robust (as can be seen from the confocal pictures in Figure 2 (the second image panel in Figure 2B was added to directly enable visualization of import for both the small and large MS2 capsids. While total brightness is different, the import signal significantly increases over time for both capsids) and Figure 3 our data show a consistent and robust trend of NLS-dependent import (see kinetic curves in Figure 3B, bottom right). Only because the data for the different capsids are all plotted together on one plot, this appears very low, relative to the much more efficient smaller capsids. We would argue that this rather unexpected large difference between cargoes efficiencies adds to the significance of our work. The new main text Figure 5B and C shows the data normalized. Here it is now clearly visible that import occurs also for the large MS2, which is also explained very well by our new simple biophysical model.

In respect to the “misleading abstract” comment, we argue that the results of our HBV experiments are not related to the technical difficulties but rather represent valid experimental data of a large cargo model (HBV with engineered NLS signals). “No import” is also a result, in particular if one puts this in the context of the widely used statement that the largest physiological import cargo known is HBV. We do agree, that for the “No import” conditions we only show cumulative indirect evidence, as we observed this for three completely different designed HBV capsids. However, as frequently with experimental studies, this refers to the tested conditions, and we discuss this now more rigorously.

The NLSs are randomly attached to the capsid surfaces, making the resultant populations heterogeneous. The quantified number of NLSs is an average, and gel analysis is semi-quantitative. Some discussion of their expected errors is warranted.

We agree with the reviewer that the gel quantification of the number of NLSs represents an average of the labelling reaction (that is expected to be Gaussian distributed). In the revised manuscript we have made sure to clarify this. It is a bulk experiment indeed, and we have no single molecule resolution to look beyond the ensemble average, which however does not make the result less quantitative. We see a very clear band shift when adding the NLS to the protein. As the NLS itself is small, the Coomassie band intensities where determined after scanning the Coomassie stained gel in a rigorous area analysis of the two different bands. Figure 1C shows clearly two nicely separated bands for NLS vs no NLS labelled capsids. The gels show a high purity, and the analysis is very robust and thus yields directly a clear well reproducible measurements of the labelling ratio. While Coomassie analysis might appear as a basic, not very sophisticated tool, its result are robust, and well quantifiable. Following the reviewer’s suggestion, we have expanded our discussion on the number of NLS determination in the Materials and methods section to explain ourselves better.

The only biologically relevant capsid (HBV) – i.e., one that is imported into the nucleus – is not included in their analysis. Physiologically, this capsid disassembles in the nuclear basket. They have deleted the authentic NLS for their experiments. Thus, the biological implications are limited.

Our goal in this work was to study general nuclear import properties using large model cargoes. Very little is known about the biophysics of large cargo transport and we think this is mainly due because most physiological cargoes do no enable to study large cargo import reliably. Taking HBV an example, the fact that the NLS is phosphorylation-dependent and potentially also packing-dependent, and import might actually require capsid disassembly, has left us with much ambiguity in the literature. In contrast, we feel that our reductionist approach provides a very good route to understanding fundamental principles. Additionally, we note that transport of HBV has been shown to depend on the same importinα/β that binds the NLS we use, so our simplified and more controllable system can still be insightful. We hope that reviewer 1 might also get persuaded by the view of reviewer 2, who specifically comments favourably on the choice of system by writing,” The effort is commendable and the use of modified viral capsids is admirably clever.”

In Figure 1D, are the curves actual data or fits? FCS and DLS signals will be dominated by the large particles, yet free dye and/or labeled capsid monomers can significantly influence the import curves – are these responsible for the non-zero ordinate intercepts in Figure 3—figure supplement 1? Can the labeled capsids be separated from monomers and free dye by size exclusion chromatography?

Figure 1D shows representative FCS data traces for the measured capsids, which we then fitted in Symphotime with a diffusion model. As common in FCS traces, very little noise is indeed visible at short correlation times. We agree with the reviewer concern about potential contaminations, and we have made a substantial effort to eliminate this concern. In short, we have performed the following:

1) Our labelling strategy makes it impossible that monomer import could be detected. As we label with dye and NLS separately, only larger structures can be both fluorescently labelled and NLS labelled. The larger the assembly gets, the more likely it would be to pick up contaminations by our analytical pipeline (FCS, DLS and EM).

2) Experimentally, the capsids are separated from the excess of unreacted dye by size exclusion chromatography and an additional step of tangential flow filtration. The sample undergoes extensive washing during this process, that should efficiently separate few kDa monomers and MDa-sized capsids and we also point out that we do not observe any capsid fragments in the final EM images. Further extensive characterization of each sample with EM, FCS and DLS lead us to conclude that the bulk of our preparation is formed by intact capsids. Please note, that FCS and DLS are sensitive to pick up substantial contaminations. As stated above (Figure 1D), also our FCS data are fit well with a single species.

3) For comparison, import of very small cargoes follows completely different kinetics (rather seconds, see e.g. Timney et el., 2006 and Ribbeck and Gorlich, 2001). This eliminates concerns of contamination of our kinetics on the relevant timescale. However, very large assemblies (like a half capsid) could easily contaminate our result. However, those we would easily detect using EM, and we were very stringent on our EM data quality, proceeding only when the EM showed very uniform nearly only perfect round structures.

We now discuss this better in the Materials and methods section and thank the reviewer for pointing out that this part was not made sufficiently clear.

The authors discount the importance of surface properties at numerous locations throughout the text. But they have not actually tested this, and surface properties are in fact surprisingly important, as multiple studies have shown – changing a few residues or adding fluorescent dyes can dramatically change the import properties of cargos. In fact, I would not be surprised if varying the number of dyes on their cargos would alter the slopes of the plots in Figure 5, or some of the scatter in these plots arises from the dye:cargo:NLS ratio. Minimally, they should tone down their discussion arguing against a minimal influence of surface properties.

While we agree with the reviewer that for most (especially passive) cargoes surface properties are key to nuclear import rates, our data point to a minor role for import of large cargoes bearing multiple NLSs. We have substantially extended our discussion about this point (see Discussion section in the main text), where we now elaborate on how much the Importin complex ends up covering (shielding) the capsid surface, even from a plain geometrical point of view.

More importantly, to address the reviewers points, we have performed new experiments, where we directly compare the MS2_S37P capsids labelled with NLS peptide and with a modified peptide bringing additional 5 charges per peptide: as can be seen in Figure 5C, the samples with/without additional charges behaved very similarly in the import assays (compare red dots and squares). This is consistent with our view that the highly decorated cargo surface is mostly shielded by bound impA and impB.

While the authors limit the fitting "to the first 40 minutes to extract more accurate kinetics", the opposite is in fact true. Accurate fitting of exponential kinetics requires knowing the asymptotic limit, which is not the case for numerous curves in Figure 3 —figure supplement 1. Also, initial time points in these curves vary widely – this is not expected or discussed.

We thank the referee for this point, and we apologize for causing this confusion. As we explain in our “major revisions summary box” at the top of this point-by-point, we now address this topic much more rigorously. In the revised manuscript, we focus our analysis on the initial influx rate, which (unlike the long term steady state) is determined by the kinetics of translocation of capsids through the pore. We find that this rate is more reliably extracted by fitting the data with a mono-exponential function (up to 80min) than a linear fit to the first few data points. We do this with an understanding that the long term steady state can be influenced by many additional factors, and make no conclusions from parameters of our monoexponential fits regarding the final steady state values – please also see the response to reviewer 2. To confirm that the mono-exponential function is a good approximation at short times, we have explored the option of a double exponential fit, and found that this does not increase the quality of the fit. Moreover, the fit is “sloppy” – whereby many combinations of the parameters produce fits of comparable quality (as reflected in the extremely high uncertainties in fit double exponential parameter values in Supplementary file 2). We further point out that using a double-exponential function to extract the initial rate gives essentially the same values as our mono-exponential fits (Supplementary file 2, Table 2). We hope these points are clear in the revised version and address the referee’s concerns.

Due to technical reasons, our first measurement point is 2 min delayed after the addition of the capsids, and therefore capsids with different properties accumulate to slightly different levels by that time – this could be a source of scatter of the “initial value”, together with minimal different levels of unspecific background signal. Thus, in the initial minutes, a potentially complex combination of capsid diffusing into the cells, non-specifically adhering to the coverslip or cell membranes, and being recruited to the nuclear envelope could give rise to different initial fluorescence. We have revised the text to clarify this point.

We also want to note, that we tested our new analysis pipeline for robustness of fitting 80 min vs 40 min, and gratifyingly the results are consistent across both fitting ranges.

For 80 min time points, the authors should really consider including CAS, RanGAP and RanBP1 to maintain complete recycling of transport factors.

Control experiments including CAS, excess GTP, excess Importinα and absence vs presence of an energy regenerating system in Figure 2—figure supplement 1 all show that the overall saturation behaviour is conserved also in those conditions. Most notably, we address this concern by a new experiment: we have now performed in vivo experiments where we microinjected live oocytes from starfish with the same representative capsid samples. The in vivo experiments agree qualitatively well with our permeabilised cell data in support with all our conclusions (see new Figure 2—figure supplement 2). Note, that the in vivo experiments areextremely low throughput and simply much harder to quantify (oil droplets, sticking, spatial heterogeneity etc., see Figure 2—figure supplement 2 for more details). The permeabilised cell assays (while still not easy when working with very large capsid cargoes) suffer much less from those problems.

"Normalized nuclear intensity" needs some explanation. Relative to what? Do these correspond to the same scale for different plots. What does an intensity of 1 signify? How does this relate to the intensity in Figure 1D? The efficiency of nuclear uptake of the different cargos varies widely, but this is not discussed.

We have expanded the discussion on the data analysis to better explain how we derive the corrected nuclear intensities to compare them between different capsids. Briefly, the raw intensity values (such as the traces in Figure 2D) are background-subtracted and corrected for the individual capsid brightness (estimated from FCS). These traces are then fitted and analysed. For display purposes, we have plotted the kinetic curved subtracted by offset parameter A in Figure 2.

The energetic discussion in the last paragraph has little meaning without an estimate of the entropic cost of displacing the permeability barrier.

In our revised manuscript we have now completely revised the biophysical analysis of the implications of our results. In collaboration with the Zilman lab (now included as co-authors on the paper), we have analysed the data through the lens of a minimal theoretical model based on previous experimental and theoretical results that incorporates basic physical insights into the kinetics and the energetics of capsid translocation. In particular, the model enables us to estimate the insertion free energy of a capsid into the NPC. The model captures our experimentally measured rates of import very well, as can be seen in Figure 5C.

Reviewer #2:The manuscript by Paci and Lemke describes experiments addressing nuclear accumulation of large NLS-labeled cargoes. The effort is commendable and the use of modified viral capsids is admirably clever. However, I have some serious problems with the interpretation.

We thank the reviewer for his/her feedback on our work. Following up on all reviewers’ suggestions, we have substantially reworked the manuscript and expanded several points with new supporting data (both experimental and theoretical). We have summarized the main changes in our “major revisions summary” at the beginning of this point-by-point, and kindly ask this reviewer to read this one before delving into the detailed point-by-point discussion below.

The experiments are based on permeabilized cell assays. These are standard in the field, for better or worse, but they suffer a generic problem in that the rest of the cell is washed away. In a live cell, the transport substrate of interest has to compete with the rest of the proteome for attentions of the transport receptors. This can have a dramatic effect on the transport kinetics.

We agree with the reviewer that the permeabilized cell assay might not recapitulate all aspects of a living cell. To further support our results, we have now added an experimental dataset looking at the import of our large cargoes into microinjected living starfish oocytes (Figure 2—figure supplement 2). These experiments are extremely labour-intensive and low-throughput compared to the permeabilized cell assays, therefore we could not aim for the same level of characterization across tens of sample. On top of that, technical difficulties make them less quantitative then the very established permeabilized cell assay, which also serves here to compare our results to the standard in the field. However, we were able to test representative samples from the three main cargoes analysed in the paper and we were able confirm that their behaviour in intact live cells matches what we observe in permeabilized cell assays (see new Figure 2—figure supplement 2).

Like most studies of nuclear accumulation, the analysis does not distinguish properly between permeability of the nuclear envelope and the saturating level of nuclear concentration. The latter is recognized as "robust nuclear import" but depends, quite obviously, on the RanGTP system. The assumption that monoexponential (first-order) kinetics measure permeability through the nuclear pores is simply not justified. The observed kinetics reflect the rate-limiting step, which may be Ran recharging with GTP or recycling to the cytoplasm. See Kim and Elbaum, 2013, and much earlier Smith et al., 2002.

We thank the referee for this excellent point and we regret that it was not clear in the previous version. Indeed, the eventual steady state nuclear accumulation levels are affected by many factors such the availability of NTRs, Ran, Rcc1 and other factors. In particular, the theoretical work by Elbaum and Kim shows the possibility of non-monoexponential approach to the steady state, where the influx of cargoes is balanced by the outflux.

To address the reviewer’s concerns, our analysis in the revised manuscript is now focused on the initial rate of nuclear import (see new Figure 5), which is independent of the long time shape of the intensity curve; we stress that we rely on the mono-exponential fit only to extract the values of the initial import rates (these fits are shown in Figure 3—figure supplement 1). In this initial stage, the increase in nuclear fluorescence is expected to be solely due to the nuclear import of the capsids, while the kinetics of cargo dissociation from transport receptors and further steps in the Ran cycle should only have a minimal effect.

Furthermore, we have re-investigated the appropriateness of the mono-exponential fit for these short time kinetics, and find that it is an excellent approximation in the regime of interest. Double exponential fits do not provide an appreciable improvement in the fit quality; on the other hand, they are “sloppy” – whereby many combinations of the parameters produce fit of comparable quality (as reflected in the extremely high uncertainties in the parameter values for the double exponential fit (new Supplementary file 2), and do not produce very different estimates of the initial import rates (new Supplementary file 2, Table 2). Finally, we also note that empirically the mono-exponential import kinetics is a very common observation in the literature including in the experimental work of Kopito and Elbaum, 2007.

Quantitative measurements of nuclear accumulation can be affected in addition by binding to structures within the nucleus, as suggested by the images in Figure 3 for MS2 with high NLS count. Each NLS adds a considerable amount of positive charge. This may well affect binding to nucleic acids when present in such high local concentration on the viral capsid, especially if DNA/RNA binding proteins are lost in the permeabilization.

We agree with the reviewer that cargo sequestration by binding to nuclear structures may affect nuclear import rates; however, the new experimental dataset we have added employing charged NLS peptides (Figure 5) supports a picture where this contribution is not substantial. In addition, our improved analysis focuses on the initial import rate, which will be minimally affected by charge accumulation.

The text deals with the level of nuclear accumulation ("endpoint" in Figure 5), but the graphs presented show the accumulation kinetics rather than the saturation as a function of #NLS. The time for half-saturation, (I(t) – A)/Imax = 1/2, is actually ln2/k, not ln2/Imax as written in the text (subsection “Image and data analysis”). Looking at the table in Supplementary file 1, the values for T_1/2 are listed equal to 1/2 * ln2/k. This has the correct units but I don't understand the factor of 1/2.

The reviewer is correct, and we indeed did extract the half-saturation as ln2/k (see supplementary file 1). We have expanded our discussion to better clarify the information extracted from the data and, also following the recommendations from all reviewers, we have now the quantitative analysis and the combination with the biophysical model on the initial transport rate, which can be extrapolated from the fitting parameters and is much more robust, as discussed above.

If the aim of the exercise is to study the degree of accumulation, i.e., Imax, then the proper parameter to measure is the saturating nuclear to cytoplasmic ratio N:C. The logarithm of this ratio is the chemical potential difference, which is the essential thermodynamic quantity. As presented, the data do not show the cytoplasmic intensity and the background correction that was applied is not described. Figure 2C shows a single example of the cytoplasmic intensity where the nuclear to cytoplasmic ratio saturates at about 10 (700 / 70 units on the graph).

As mentioned above and following the reviewer suggestion, we now instead focus our analysis on the initial import rate, which isolates the step of translocation through the NPC from the efflux that can be affected by multiple factors. We agree with the reviewer that the N:C ratio could potentially capture the transport efficiency of the different cargoes but we have to note that this quantity is unfortunately not robust enough in the case of our sample to enable a thorough comparison of all samples. This is because some of the capsids have unspecific interactions with cytoplasmic structures and membranes (see for example Figure 3A, I53-47 in absence of NLS) that prevent us from extracting a clean and robust cytoplasmic signal from all samples.

Since the fluorescence external to the cells coming from titrated cargo substrates should equilibrate with the fluorescence in the cytoplasm, I looked to see if this might be included in the fitting parameter A. It was not clear whether A is the background correction itself or a fit after the correction is applied. In any case A cannot represent the fluorescence from free cargo. According to the text these are introduced at a constant 8 nM concentration, but the values listed in the supplementary file vary widely, even for a given class of cargo. Why should they vary so widely? Presumably these values are corrected by the same factor as Imax for the substrate brightness. If they are not corrected, shouldn't the capsids with fewer NLS appear brighter, so with larger A? In some cases, A is a very large fraction of Imax, leaving little dynamic range for the measurement itself. (Compare I53-47 with 15, 18, and 22 NLS.) In principle the black level to subtract is that of the confocal microscope with the laser blocked, and the fluorescence in the surrounding medium should match that measured in the permeabilized cytoplasm. If the cells are auto-fluorescent in the measurement channel then some additional correction will be required, but it should be specified clearly.

A is the offset parameter used in the fitting of the corrected kinetic traces. All traces have been corrected for cargo brightness (estimated from FCS) and PMT background of the microscope prior to fitting. Due to technical reasons, our first measurement point is 2 min delayed after the addition of the capsids, and therefore capsids with different properties accumulate to slightly different levels by that time. In those initial minutes, a potentially complex combination of capsid diffusing into the cells, non-specifically adhering to the coverslip or cellular membranes, and being recruited to the nuclear envelope could give rise to different initial fluorescence. We have revised the text to clarify this point.

A few relatively technical points:Why was the labeling with fluorescent dye and NLS done both on cysteine? The proteins could have been labeled first on lysine and then with NLS on the cysteine. The problem is that the molecular weight of the dye is almost half that of the peptide. Is a control available to show that the dye labeling really has no effect on the gel mobility? Figure 1—figure supplement 1 shows both Coomassie and fluorescence in the "unsuccessful" labeling of I53-50. For clarity, the main figure should also show the fluorescence in the successful case.

Following the reviewer’s suggestion, we have added an image of the fluorescence signal in the labelled samples in Figure 1C. In control samples without NLS we have verified that there is no substantial effect on gel mobility (a clear single band is consistently observed). Furthermore, the NLS includes many Ks, and many of those are of functional relevance: the cysteine labelling strategy ensures that we do not impair functionality, and that at most we add one label per monomer subunit. Having two labelling sites on one monomer could introduce problems with contamination of kinetic signals due t very small capsids, (see also our reply to reviewer 1 regarding the point of contamination due to broken capsids).

I did not understand the toy model in subsection “Global quantitative analysis of nuclear import in relation to cargo size and #NLSs”. The binding energy of NTRs to the cargo does not assist in directional translocation, nor is it transferred to displacing the FG repeats. That depends on interactions of NTRs with FG motifs. Crowding in the nuclear pore as shown in Figure 5 is interesting and might relate to kinetics, but not to the saturating concentration ("endpoint").

In our revised manuscript we have completely reworked (see box at the beginning of this pbyp) this part by analysing the data using a minimal biophysical model based on previous examinations of the in vitro experiments of the thermodynamic permeability of the FG-Nup assemblies, and kinetics of transport through NPC mimicking nanopores and NPCs in permeabilized cells. The model allows us to examine the major contributions to the transport kinetics: free energy cost of insertion of a capsid the NPC and the energetic gain from interactions between NTRs and FG motifs. The model and the corresponding analysis are now described in subsection “Quantitative analysis of the nuclear import in relation to cargo size and #NLSs” and new Figure 5C.

Nuclear export is not just the inverse of import. See Kim and Elbaum, 2013. There is a fundamental difference between exchange of RanGTP, a reversible reaction in "import", and physiologically irreversible GTP hydrolysis, which is coupled to translocation in "export".

We have now revised the according part of the text to better reflect the differences between the import and the export mechanisms.

The manuscript is long for a short report, about 3500 words in the main text alone.

Following a further extension of our work with new data and the theoretical model, we have now changed the format to a research article.

Hoping to end on a constructive note, I have to apologize for being such an ornery reviewer here. I do quite like the experiment and I believe the data hold some new truths to be discovered. Wherever the work is ultimately published, I would like very much to see the nuclear accumulation presented as the nuclear to cytoplasmic ratio. This will normalize inherently for substrate brightness and avoid potential inconsistencies carried in by numbers from other measurements, imprecise dilutions, protein losses in aggregation, etc. Surely the data are available without requiring any further experiments. I am sure they could be reanalysed easily, avoiding confusion between kinetics and saturation. Plotting the ratio will clarify whether the additional number of NLS indeed influence the kinetics and saturation as suggested. There might be surprises in store.

We thank the reviewer for his constructive criticism and for adding this encouraging and motivating note. We agree with the reviewer, that a nuclear to cytoplasmic ratio analysis would have been nice, and this is what we originally tried. However, due to the different stickiness of the large capsid structures to cellular structures, the nuclear to cytoplasmic ratio does not yield robust results. In fact, as written in the main text, these capsid structures are already the best we identified from a much larger set of “big structures” we tried to get workable for these experiments.

We are convinced though, that our much improved analysis pipeline and completely revised fitting strategy deals well with all issues pointed out the by the reviewers, so that we hope the paper can now proceed to publication.

[Editors’ note: what follows is the authors’ response to the second round of review.]

One of the original reviewers feels the manuscript has been improved but has some issues with the interpretation of the data, and the model. Specifically, the reviewer states "Particular attention must be made to predictions of the model, and interpretations in the context of this model." This reviewer has been thorough in the evaluation, so we feel the comments may likely be helpful in improving the manuscript further.Because the concerns can be answered without additional data, but only require revisions to the manuscript, or explanations for the reviewer, we opt to send it back to you to address these comments.Reviewer 1:This revised manuscript has been substantially improved by tightening up the discussion and presentation to focus on the main story, and with the addition of a mathematical model. However, I do have some concerns about the revised manuscript, listed below in order of importance. While some of these points address accuracy and a logical consistency, other portions are intended to promote a more nuanced and informative picture. Particular attention must be made to predictions of the model, and interpretations in the context of this model.

We thank the reviewer for his constructive feedback on our work.

1) Figure 5B – The model impressively explains the values in the graph. However, all of the ∆G values are positive, suggesting that binding to the permeability barrier is unfavorable. Nonetheless, nuclear rimming is clearly seen during the import experiments, indicating that interaction with the pore is favorable – more favorable than being in the cytoplasmic compartment. This indicates that the NPC is a thermodynamic sink. The data thus seem incongruent with the model, which only postulates an energy barrier. The model in Figure 5—figure supplement 4 is reminiscent of the vestibule model of Tu et al., 2013), yet here too, none of the ∆G values are negative (which was the case in Tu et al.). Please discuss.

This is an excellent question, and we regret this wasn’t explained clearly in the previous version. Nuclear rimming without substantial nuclear accumulation suggests that capsids interact favourably with some regions of the NPC, however not necessarily all regions, as otherwise nuclear accumulation would be visible.

In this respect, a variant of the vestibule picture provides one potential resolution of this apparent contradiction. In a spatially non-uniform effective potential inside the NPC, the averaged *ΔG* is dominated by regions of the potential that have a higher (more positive) value, while the cargoes still accumulate in the regions with low local *ΔG*. It may be a thermodynamic sink with locally negative free energy even while the global *ΔG* is positive.

In the case of the NPC, the most likely explanation for the “rimming” is a region of relatively low density – a diffuse “cloud” of FG nup density at the cytoplasmic side with very low penetration cost. This region has been noticed both in Tu et al., (as a “cytoplasmic vestibule”) and in Lowe et al., 2010 as cytoplasmic “docking” region.

To address the referee’s concern, we have now explicitly incorporated the cytoplasmic “docking/vestibule” area into the free energy profile, as shown in Figure 5—figure supplement 4. Note that the presence of such vestibule affects the flux through the pore minimally because the pore selectivity properties are mostly dictated by the barrier region.

We agree with the referee that the detailed comparison between different models is important. However, we emphasize that it is impossible to make unambiguous resolution of these question based on the bulk measurements only. The current minimal model that is warranted by the data is offering guidelines for the data interpretation, and we relegate more detailed models to the future work with single molecule tracking.

2) They cite four references for the initial flux equation (2, 33-35). I cannot find the equation they use in these references. In fact, two of them describe flux in terms of a constant multiplied by a concentration difference, which seems inconsistent with their equation. More discussion is necessary to elucidate where the model comes from.

We apologise for confusing references. We have updated these, and expanded the explanation of the formula in the main text. We also note that in the case of (idealized) non-interacting cargoes, the flux from the cytoplasm to the nucleus is J→=kONcCa+eΔG, the reverse flux is J→=kONcNa+eΔG (without taking into account Ran complications) and the overall flux through the pore is J→−J←=kON(cC−cN)a+eΔG, so all these formulations are consistent with each other (in the regime of the approximation validity).

3) If I understand the methods correctly, the NLSs and dyes were simultaneously mixed with the capsids. They discuss tuning the NLS/capsid ratio, and this is ultimately determined via a gel shift assay. But what about the number of dyes per capsid? It seems like they have brightness data from FCS experiments, and this should be reported. Do the number of dyes vary inversely with the number of NLSs? They continue to minimize the role of surface properties, yet a few extra dye molecules were shown by Tu et al., to dramatically affect the permeability properties of the cargo. I do not consider it safe to assume that the number of dye molecules does not influence the particle's interaction strength with the NPC.

In our experimental pipeline, we quantify the number of dyes per capsid via FCS: following the reviewer suggestion we have now included the capsid brightness values to Table 1. Throughout our whole dataset we don’t find a strong correlation between NLS and dye number (R^2^=0.22). Similarly, we don’t see a correlation of the labelling ratio with the initial import flux (R^2^=0.14). In the case of the cargo used in the Tu et al. study, the ΒGal enzymatic activity (disaccharide hydrolase, potentially binding disaccharides in the NPC) also contributed to the unspecific import, requiring a total of 4 mutations to obtain a specific import cargo. It is possible that the drastic effect of adding 8 vs 16 dyes is somewhat specific to the ΒGal cargo and its surface, as the same group previously reported that the presence of 4 Alexa 647 dyes did not affect importinα-2XGFP-NLS interaction and cargo residence time (Sun et al., 2008). Our cargoes don’t present complications as the enzymatic activity of ΒGal and for the 0 NLS case we clearly don’t see any import even in presence of high dye numbers (up to 23 for the MS2S37P cargo, most comparable to the ΒGal) and after waiting much longer time (1.5-2 hours).

Furthermore, the shape of the ΒGal is very different compared to our spherical capsids (cylindrical with only 9 nm in diameter for the smaller axis), which in relative comparison to our study, render the ΒGal a small(er) cargo. Following the recommendation of the reviewer, see also below, we discuss now much more carefully in the discussion the surface shielding effect we speculate about for our large spherical structures due to Importin coverage. Those might not be relevant to ΒGal with only for Importins. Also, the topic of dye labelling is now discussed more carefully.

Moreover, they state that F(R) scales with the radius, yet the values for F(R) that they obtain are all essentially the same, which would be consistent with different surface properties of the different diameter capsids. Stating this does not diminish their results.

This is an excellent point, and we regret again that it was not clear in the previous version. The theoretical (and sensible) expectations are that the insertion cost should increase with the capsid size. However, the specific predictions have been obtained only for particle sizes much smaller than the ones studied in this paper, and only serve to guide the assumptions of the model.

One potential technical reason for the similarity of the observed values of F(R) is that the flux of the capsids without NTRs is so low that it is beyond the noise threshold of experimental detection, resulting in similar values of F(R). To control for this possibility, we re-did the analysis, now excluding the zero NLS point from the model fit – and the results of the analysis have not changed significantly. This is now included in the Figure 5—figure supplement 5.

Other potential reasons are that for such large capsids resulting in extreme compression of the FG nups, the insertion cost essentially saturates to its maximal value. We have modified the text accordingly to reflect this point.

Finally, we agree with the referee that at the moment we cannot exclude the possibility that at least at very low NTR coverage (where we basically cannot reliably detect any flux experimentally), surface effects might play a role. We have also added this to the revised discussion.

4) The epsilon values are surprisingly small. For the cargo of Tu et al., this would predict a very small interaction strength of the fully decorated cargo, and even smaller for a single NTR-bound cargo, which nonetheless still clearly binds to the pore. Note that the size (volume occupied) of β-galactosidase is less than MS2(S37P) by a similar ratio that the MS2(S37P) size is less than I53-47. It would be quite surprising indeed if the substantial behavioral differences of the β-galactosidase and MS2(S37P) cargos can be ascribed to the size and shape differences alone. It seems that surface properties must at least somewhat contribute to the observed differences.

We thank the referee for this question. We note that there are two separate issues here: (1) the model-independent experimental fact that even the smallest MS2(S37P) capsid requires significantly more bound NTRs than theΒGal cargo used in Tu et al., and (2) the small (5-6 times smaller than in Tu et al.,) values of the effective binding energy ϵ per NTR in the model that reflect the experimental fact 1.

It is hard to make detailed quantitative comparisons between different experimental platforms, given inherent uncertainties in experimental conditions. However, we feel that the apparent discrepancies between this work and Tu et al., can stem from the following factors.

The ΒGal construct of Tu et al., is a roughly cylinder shaped molecule of 18 nm at its longest axis, similar to the MS2^S37P^. However, its smaller axis is 9 nm with an accordingly lower cost of insertion, which can explain well why for this substrate 4 NLSs might be enough for import.

We would like to clarify that the ϵ values in Table 1 are physically realistic. The value of the effective energy ϵ per NTR is a product of the bare energy ϵ0 and the average volume fraction ϕ of FG motifs inside the pore. Using common estimates of the FG motif numbers in the pore of the order of ϕ≃0.01 , the bare interaction energy between one NTR and one FG motif is on the order of ϵ0≃3−12kBT  (and up to 15kBT in the vestibule model), in accord with Tu et al. and other measurements NTR-FG interactions (for instance, by the Lim group). We have updated the manuscript to clarify this point further.

We cannot exclude at this point that the differences between the surface properties of our capsids and the ΒGal construct might contribute to the observed differences at low NTR coverage – and we now modified the text accordingly. It is also possible that at low NTR our values of ϵ could be closer to those of Tu et al. However, they are unlikely to be the dominant explanation at high NTR coverage, which is the predominant focus of this work. At high NTR coverage, potential sources for the discrepancy could be lower availability of the FG motifs due to extreme compression of the FG nups and/or competition between the FG motifs for the NTRs on the capsid.

We have clarified these issues in the text, and hope in the current form the manuscript addresses the reviewer’s concerns.

Finally, cargo shape has been observed to impact transport properties, with passive elongated cargoes being transported faster than spherical one of the same size (Mohr et al., 2009) and this is likely even stronger in the case of large cargoes, for example considering baculoviruses that need to specifically orient along their long axis to enter through the NPC.

5) Discussion section – I do not understand these surface coverage calculations. For maximum NLSs of 38, 35, and 98 for MS2(S37P), I53-47, and MS2, I get 84%, 42%, and 85% surface coverage assuming 20 nm^2^/β.

For the surface calculations we have taken the samples with highest NLS numbers, respectively 54, 44 and 98 for the three capsids (see Table 1). This explains the difference in the calculated values, and we have now clarified this better in the text.

This does not include Importin α. How much do the diameters increase assuming a full coat of Importins α and β? This is expected to be significant. How does this increased diameter compare with the size of the channel?

As the reviewer correctly points out, for simplicity we haven’t considered importin α explicitly in our back-of-the-envelope estimates of surface coverage. Taking the globular conformation of bound importin β, we could expect a radial increase of 9 nm in case of 100% cargo surface decoration by NTRs. The import complexes would then have an overall diameter of 34, 40 and 44 reaching a size comparable to the scaffold channel itself (approximately 40 nm).

However, we would like to stress that these size increase estimates are in our view not definitely meaningful. If one considers the increasing evidence that importins are an integral part of the permeability barrier (as for example reviewed in Lim et al., 2015) and FG-Nups/NTR complexes are highly dynamic (see Milles et al., 2015 and Hough et al., 2015) so FG-Nups can easily penetrate the cargo-importin complex. In such cases the additional layer due to Importin coverage does not effectively increase the volume of the transported cargo, as this layer is part of the transport machinery and readily penetrated by Nups which do interact with the Importins.

We therefore find that it is more appropriate to focus on the bare cargo size (which we could consider the FG-Nup excluded volume), and compare results accordingly.

Is there any experimental evidence that all NLSs on the capsids are bound to NTRs? Taking into account that concentrations and the Kd (~40 nM, α for NLS) are similar, the NLSs on the MS2 capsid are only about 90% occupied, implying 77% surface coverage. While these changes may not materially change their interpretation, a more detailed discussion is necessary to build an accurate picture and to build confidence in the conclusions. Other potential complications: (1) is it possible geometrically for all NTRs on a capsid to be bound to FG repeats? Figure 5A suggests that this may not be possible;

To ensure maximum binding of cargo and NTRs, we have always used excess of importins and pre-incubated them at high concentration with the cargo prior to start of the experiment. In DLS measurements of cargo + transport mix we detect two peaks corresponding to the capsids and small proteins in the mix, if we incubate capsid and importins we only detect the capsid peak, pointing to most importins being bound. Indeed, this does not necessarily ensure complete saturation of the NLSs, but as the reviewer him/herself points out, it still does not change the results, are we are always comparing the cargoes to each other in similar conditions. As for binding of NTRs to FG repeats, we do not see how the (perhaps oversimplified) scheme in Figure 5A suggests that it isn’t possible: FG-Nups are highly flexible and can extend easily several tens of nm, so they could easily enwrap the import cargo complex. For the sake of simplicity, we have represented this in a rather simplified way in the scheme.

Of course, it is indeed possible that, at a given moment, a given NTR is not bound to an FG motif at any of the bindings sites on the NTR. However, when we did a toy model calculation in our 2015 paper for only two binding sites, we saw that the probability of being unbound drops dramatically relative to a single site binding model. With Importins and Nups being highly multivalent, this becomes very unlikely. In that respect, ϵ is the *average* binding energy of an NTR to the FG domains, which already includes this possibility.

and (2) can multiple capsids simultaneously bind to a single pore? Excess cargo, slow import and nuclear rimming suggest this possibility. Would this affect interpretation?

Although we cannot exclude multiple capsids binding to a single pore with absolute certainty – and this has indeed been reported in EM of HBV capsids injected in *Xenopus oocytes* (Pante and Kann, 2002) – simple theoretical estimates show that these are likely lower probability events under our experimental conditions, at least during the initial period important for the calculation of the initial flux, which is the focus of our analysis.

6) It is unclear whether there is any meaning behind the A values. These are highly variable, and I don't know what to make of them. In principle, A could reflect the accumulation of the cargos on the nuclear envelope, but as this arises from an extrapolation to zero time, it seems like this should in fact be zero, or at least some reasonably explained value. One possibility is that import rate could be dependent on the amount of accumulated cargo at the pores, i.e., a release rate, as entrance into the NPCs appears really fast.

We attribute the differences in A values to limitations in the experimental design: (i) The start of the experiment is given by manual pipetting of the capsid solution to the well. We always image two wells (each 8 regions) in parallel using an automated data acquisition imaging procedure. The different capsids have to diffuse into the cells, and could be recruited to the nuclear envelope at a slightly different initial rate. Slightly different cargo accumulation in the first 2 minutes of the experiment prior to the start of the experiment is intrinsic to the resolution of the manual pipetting defining the start. (ii) In addition, we have observed different amounts of non-specific adhesion of the capsid samples to cellular membranes, as can be seen for example by comparing the confocal images of MS2^S37P^ and I53-47 in Figure 3. Note that we do not extract any interpretations from this offset parameter, but simply use it to account for this complex mixture of effects giving rise to slightly different initial values. Finally, this parameter is quite sensitive to imperfections in the kinetic fit, which could be a further contributing factor to its different values.

7) The data on negatively charged linkers is inconclusive at best, as they are highly scattered. Their conclusions should be toned down.

Following the reviewer’s feedback, we have toned down our discussion of the experiments with charged linkers.

[Editors’ note: what follows is the authors’ response to the second round of review.]

The manuscript has been improved but there are some remaining issues that need to be addressed before acceptance, as outlined below:One reviewer feels the manuscript is substantially improved but there remains an outstanding issue that has not been corrected in the revision. The reviewer feels that Figure 5—figure supplement 4 needs to be clarified as described below. Additional minor comments directed at improving the manuscript are included as well. Please send a revised manuscript that addresses these comments sufficiently that it may not need to go back to this reviewer.This revised manuscript has been substantially improved, with a much more balanced and informed discussion. All of my major concerns have been adequately addressed, with the exception of one item, the model in Figure 5—figure supplement 4. The figure itself is confusing/unclear, and I do not understand the basis behind building the model the way they did. Specific concerns for this figure are as follows:1) What is the y-axis in the top panel of 'A'? This should be marked. My guess is that this is some measure of 'FG-Nup density' – are there any relevant units?

We apologize that this had been overlooked. The relevant units are the volume fraction. We have re-drawn the diagram in Figure 5—figure supplement 4 and included this information.

2) The dimensions of L1 and L0 do not reflect the values in the caption. Consequently, the diagram is misleading. The Greek letter is inconsistent with the caption. The vestibule region is not marked.

We sincerely apologize for the confusion, and made sure that in the current version the labeling and the values of l1 and l0 are consistent throughout the text and the captions. We have also re-drawn the diagram in Figure 5—figure supplement 4 to match the fonts of ϕ everywhere, and clearly marked the vestibule region.

3) It is unclear why a transition region (L1) is included between the vestibule and L0. Comparing the top and bottom panels in A, it appears that the vestibule is equivalent to the cytoplasm. This does not make sense.

We apologize for the confusion. The “vestibule” indeed represent the low density cloud of the FG nups that extends into the cytoplasm beyond the NPC scaffold region. In that sense it can indeed be seen as a variant of the “vestibule” region of Tu et al., (2013) and/or “capture” region of Lowe et al., (2015). The region of the intermediate density l1 is simply the transition region between the vestibule and the high density barrier region within the NPC. We have edited the figure and the main text to explain these issues more clearly. Please also see the reply to the next comment.

4) For L1 = 30 nm and L0 = 5 nm, the first impression is that the barrier gate is biased toward the nucleoplasmic side. Is this the intention? Such a model would be consistent with the nucleoplasmic gate hypothesized by the Weis group, and, if so, should be mentioned. Alternatively, are both the cytoplasmic and nucleoplasmic L1 regions both 30 nm? This would place the barrier in the center, but very narrow. It doesn't make much sense for a 'transition region' to be 6 times the width of the main barrier, so some discussion is needed here.

We apologize again for the confusion in the values for l1 and l0 in the figure, which have now been corrected. l0=30nm is the central high density “barrier” region, while l1=5nm is the intermediate density transition region. There are two “peripheral” l1 regions on either side of the central barrier ( l0 region), therefore the barrier is not biased and is located at the center of the NPC.

5) It is unclear why the ΔG for the L1 region changes substantially for the different viral particles, yet the ΔG for the L0 region changes minimally. It seems that the ΔG for the more dense FG nup environment would be more sensitive to particle size. An older hypothesis suggested dense clouds on the nucleoplasmic and cytoplasmic sides, but significantly lower density within the center. Is this being considered here?

This is a great question. According to the model analysis, the costs of insertion into the central “barrier” region are relatively similar for all capsid sizes. By contrast, they are more variable in the l1 region. This is the main reason why the values of ΔG  in the l0 region are more similar between the capsids of different sizes than in the l1 region.

One potential mechanistic explanation is that the insertion cost into the dense l0 region is probably close to saturation already even for the smallest capsid, while in the less dense l1 region, the difference between the capsids is more visible. However, we emphasize that the model depicted in the Figure 5—figure supplement 4 is just one possible model of the NPC informed by and consistent with the previously suggested ones. Full examination of different models and the inference of the actual density distribution of the FG nups inside the NPC is beyond the scope of the present paper, and will be studied in future work.

6) The authors are correct in their rebuttal that only a portion of the NPC needs to contain a region where the interaction free energy is negative, in order to be consistent with the experimental observation of rimming. However, none of the regions illustrated in Figure 5—figure supplement 4 have negative ΔG. There is a dashed region that is apparently of negative free energy, but what this is remains unclear (point 3), and it is not clear if this energy is included in any way in their fit to the data.

We thank the reviewer for this comment. Indeed, the free energy in the “vestibule” shown by the dashed line is negative and is responsible for the observed “rimming”. We have modified our schematic and the descriptions in the main text and figure caption to clarify this issue.

7) In the lower panel of B, the green curve fit approximates the data very poorly, but does much better in the upper panel. Something seems amiss here.

We thank the reviewer for this observation. This was indeed an oversight on our side, where the values of the parameter aRan  were mixed up between the lower andthe upper panels. We sincerely apologize for the mistake and have corrected the figure; the quality of fit in both panels is now consistent.